# TORC2-dependent protein kinase Ypk1 phosphorylates ceramide synthase to stimulate synthesis of complex sphingolipids

Alexander Muir[1,2], Subramaniam Ramachandran[1], Françoise M Roelants[1], Garrett Timmons[3], Jeremy Thorner[1]*

[1]Division of Biochemistry, Biophysics and Structural Biology, Department of Molecular and Cell Biology, University of California, Berkeley, Berkeley, United States; [2]Chemical Biology Graduate Program, University of California, Berkeley, Berkeley, United States; [3]Department of Chemistry, University of California, Berkeley, Berkeley, United States

**Abstract** Plasma membrane lipid composition must be maintained during growth and under environmental insult. In yeast, signaling mediated by TOR Complex 2 (TORC2)-dependent protein kinase Ypk1 controls lipid abundance and distribution in response to membrane stress. Ypk1, among other actions, alleviates negative regulation of L-serine:palmitoyl-CoA acyltransferase, upregulating production of long-chain base precursors to sphingolipids. To explore other roles for TORC2-Ypk1 signaling in membrane homeostasis, we devised a three-tiered genome-wide screen to identify additional Ypk1 substrates, which pinpointed both catalytic subunits of the ceramide synthase complex. Ypk1-dependent phosphorylation of both proteins increased upon either sphingolipid depletion or heat shock and was important for cell survival. Sphingolipidomics, other biochemical measurements and genetic analysis demonstrated that these modifications of ceramide synthase increased its specific activity and stimulated channeling of long-chain base precursors into sphingolipid end-products. Control at this branch point also prevents accumulation of intermediates that could compromise cell growth by stimulating autophagy.

**\*For correspondence:** jthorner@berkeley.edu

**Competing interests:** The authors declare that no competing interests exist.

## Introduction

A eukaryotic plasma membrane (PM) is a complex structure composed of many protein (*Sachs and Engelman, 2006*) and lipid (*Simons and Sampaio, 2011*) species arranged with a high degree of compositional and spatial organization. The levels and distributions of lipids are clearly important for PM processes—from solute transport (*Divito and Amara, 2009*), to endocytosis (*Platta and Stenmark, 2011*), to receptor function and signal transduction (*Groves and Kuriyan, 2010*). It is essential, therefore, for a cell to maintain proper lipid balance. How cells sense alterations in PM organization in response to developmental cues or environmental stresses and adjust the rates of the reactions that maintain lipid homeostasis are vital mechanisms to understand.

We (*Roelants et al., 2011*) and others (*Berchtold et al., 2012*; *Niles and Powers, 2012*) have shown that dynamic changes in the function of Target of Rapamycin (TOR) Complex 2 (TORC2) and in a downstream protein kinase that it stimulates, Ypk1 (and its paralog Ypk2), are important for cell survival in response to membrane stress. Ypk1 (and Ypk2) are two members of the AGC kinase family (*Pearce et al., 2010*) and orthologs of mammalian SGK1 (*Casamayor et al., 1999*). In *Saccharomyces cerevisiae*, Ypk1 (and Ypk2) are activated at a basal level by phosphorylation of a Thr in the activation

**eLife digest** Cells are enclosed by a plasma membrane that separates and protects each cell from its environment. These membranes are made of a variety of proteins and fatty molecules called lipids, which are carefully organized throughout the membrane. When cells experience stresses such as heat or excessive pressure, the plasma membrane changes to help protect the cell. In particular, more of a group of lipids called sphingolipids are incorporated into the membrane under stress conditions.

In yeast cells, a protein called Ypk1 plays an important role in protecting the cell from stress. Ypk1 controls the activity of a number of proteins that are responsible for balancing the amounts of different types of lipids in cell membranes. The combined action of these Ypk1-dependent proteins leads to the remodelling of the cell membrane to protect against stress. While several proteins that work with Ypk1 are known, some of the changes that serve to protect the plasma membrane cannot be explained by the action of these proteins alone.

To provide a more comprehensive picture of how Ypk1 helps cells to respond to changes in the environment, Muir et al. developed a new approach that combines biochemical, genetic and bioinformatics techniques to survey the yeast genome for proteins that could be Ypk1 targets. Muir et al. first produced a list of potential candidate proteins by searching for proteins with features similar to known Ypk1 targets, and then considered those that are known to be involved in processes that also involve Ypk1. To filter the potential targets further, Muir et al. performed experiments in yeast cells to see which proteins prevented normal cell growth if they were over-produced. Further experiments investigating which of these proteins interact with Ypk1 when purified identified 12 new proteins that are most likely targets of the Ypk1 protein.

Two of these newly identified Ypk1 target proteins form part of an enzyme complex called ceramide synthase, which produces a family of waxy lipid molecules from which more complex sphingolipids are built. Muir et al. discovered that during stress, Ypk1 enhances the activity of the ceramide synthase enzyme, which increases lipid production and the amount of sphingolipid deposited in the cell membrane. If this process is interrupted at any stage, cells struggle to survive under stress conditions.

The other candidate proteins identified by Muir et al. remain to be validated and characterized as Ypk1 targets. Nevertheless, the techniques used have conclusively identified some new Ypk1 targets and could also be applied to similar searches for proteins targeted in other biological processes.

loop by the eisosome-associated protein kinase Pkh1 (and its paralog Pkh2) (*Casamayor et al., 1999*; *Roelants et al., 2002*). The ortholog responsible for this reaction in mammalian cells is PDK1 (*Mora et al., 2004*). However, this level of activity is not sufficient for Ypk1 to permit survival when cells are challenged with certain membrane perturbants (*Roelants et al., 2004*). Under such conditions of membrane stress, TORC2 stimulates Ypk1 and Ypk2 function by phosphorylating at least two C-terminal sites (*Kamada et al., 2005*; *Roelants et al., 2011*; *Niles et al., 2012*). Constitutively-active Ypk1 and Ypk2 alleles bypass the need for functional TORC2, indicating that Ypk1 and Ypk2 are solely responsible for executing all the essential downstream functions of TORC2 (*Kamada et al., 2005*; *Roelants et al., 2011*; *Niles et al., 2012*).

How membrane stress is communicated to TORC2 is a question of on-going research. It has been reported that membrane stress caused by inhibition of sphingolipid synthesis or membrane stretch (induced by hypotonic shock) causes two PH domain-containing proteins (Slm1 and Slm2) to relocalize from eisosomes to a separate PM region that contains TORC2 and leads to increased TORC2 activation of Ypk1 and Ypk2 (*Berchtold et al., 2012*), purportedly because Slm1 and Slm2 recruit Ypk1 and Ypk2 to TORC2 (*Niles et al., 2012*). However, other evidence indicates that Avo1 (ortholog in other organisms is Sin1) is the subunit of the TORC2 complex primarily responsible for substrate recognition of Ypk1 and its orthologs, including association of *S. cerevisiae* Ypk2 with Avo1 (*Liao and Chen, 2012*), Gad8 with Sin1 in fission yeast (*Ikeda et al., 2008*; *Kataoka et al., 2014*), and SGK1 with mSin1 in mammalian cells (*Jacinto et al., 2006*; *Yang et al., 2006*; *Lu et al., 2011*; *Liu et al., 2013*). Regardless of the actual mechanism by which the activity of the TORC2-Ypk1 signaling module is affected by

these assaults on the PM, it clearly sets in motion processes that allow the cells to cope appropriately with these stresses.

Several targets of Ypk1 have already been identified that shed light on how TORC2-Ypk1 signaling reprograms cellular processes to cope with PM stress. Ypk1 phosphorylates and negatively regulates Fpk1, a protein kinase responsible for activation of PM-localized aminophospholipid flippases (*Roelants et al., 2010*), thereby fine-tuning the phosphatidylethanolamine content of the inner leaflet of the PM bilayer. Ypk1 also phosphorylates and alleviates the inhibitory function of two endoplasmic reticulum (ER)-localized proteins (Orm1 and Orm2) that impede the function of the first enzyme unique to sphingolipid biosynthesis, L-serine:palmitoyl-CoA acyltransferase (SPT) (*Roelants et al., 2011*; *Berchtold et al., 2012*; *Sun et al., 2012*), thereby increasing the rate of formation of the long-chain base precursor (phytosphingosine) to yeast sphingolipids. Ypk1 also phosphorylates and inhibits one of the two isoforms (Gpd1) of glycerol-3-phosphate dehydrogenase, a primary source of glycerol-3P for production of glycerophospholipids (*Lee et al., 2012b*). In response to hyperosmotic shock, and in contrast to other PM stressors, TORC2 activity is markedly decreased, preventing Ypk1-mediated inhibition of Gpd1 [which also gets upregulated transcriptionally (*Ansell et al., 1997*)], thereby greatly increasing synthesis of glycerol-3P, which is dephosphorylated to produce glycerol, an innocuous osmolyte that yeast cells accumulate as a means to counteract the increase in external osmotic pressure (*Lee et al., 2012b*). Control of all these reactions already made it clear that TORC2-Ypk1 signaling is a central regulator of PM lipid homeostasis.

To gain further insight into how the TORC2-Ypk1 signaling axis contributes to PM homeostasis and potentially other cellular processes, we devised a three-step procedure to screen in an unbiased and genome-wide manner for additional candidate Ypk1 substrates whose physiological relevance we could then evaluate. As described here, we first used bioinformatics to search the yeast proteome for presumptive targets, then applied a genetic method to narrow down the hits to likely, functionally important substrates, and then used biochemical analysis both in vitro and in vivo to validate the best prospects. Although we report here the general outcomes of this screen, we focus mainly on our discovery and demonstration that both of the catalytic subunits of the ER-localized ceramide synthase complex (*Schorling et al., 2001*; *Vallée and Riezman, 2005*) are *bona fide* targets of Ypk1. These findings provide new insight into how TORC2-initiated signaling regulates flux through the sphingolipid pathway and how this specific control mechanism is important for survival when sphingolipid synthesis is compromised. In addition, we show that this regulation of the ceramide synthase complex is important for preventing accumulation of pathway intermediates that would otherwise compromise cell growth by triggering an inappropriate autophagic response.

## Results

### A three-tiered screen to identify new Ypk1 substrates

We devised a three-step strategy (*Figure 1A*) to pinpoint *bona fide* cellular targets of Ypk1, utilizing bioinformatics to predict potential Ypk1 substrates, then an in vivo genetic test involving a novel variation of the synthetic dosage lethality method to winnow the list to likely candidates, and finally biochemical analysis in vitro to confirm whether the identified gene product serves as a direct substrate of Ypk1. The physiological relevance of Ypk1-dependent modification of each protein on the resulting final list could then been evaluated.

For initial bioinformatic search of the yeast proteome, we developed a position-weighted consensus sequence logo (*Figure 1B*) for the preferred Ypk1 phospho-acceptor site based on two primary criteria: (a) the known Ypk1 sites in five, validated in vivo targets (Fpk1, Fpk2, Orm1, Orm2, and Gpd1) (*Roelants et al., 2010*, *2011*; *Lee et al., 2012b*; *Niles et al., 2012*; *Sun et al., 2012*); and, (b) the sequence preference displayed by Ypk1 for phosphorylation of synthetic peptides in vitro (*Casamayor et al., 1999*; *Mok et al., 2010*). All demonstrated substrates either in vivo or in vitro contain Arg at positions −5 and −3 with respect to the phosphorylated Ser (or Thr); thus, these positions were invariant in the search motif. Given that nearly all the verified sites within known in vivo targets possess a hydrophobic residue (V, I, F) at position +1, the search motif gave preference to sites with a hydrophobic residue at the +1 position. We then took advantage of the existing MOTIPS motif analysis package (*Lam et al., 2010*) to identify those *S. cerevisiae* gene products that contain occurrences of the search logo. Several authentic Ypk1 substrates (*e.g.,* Fpk1, Orm1, and Orm2) contain multiple Ypk1 phosphorylation sites. Thus, we filtered our search further by prioritizing candidates containing multiple

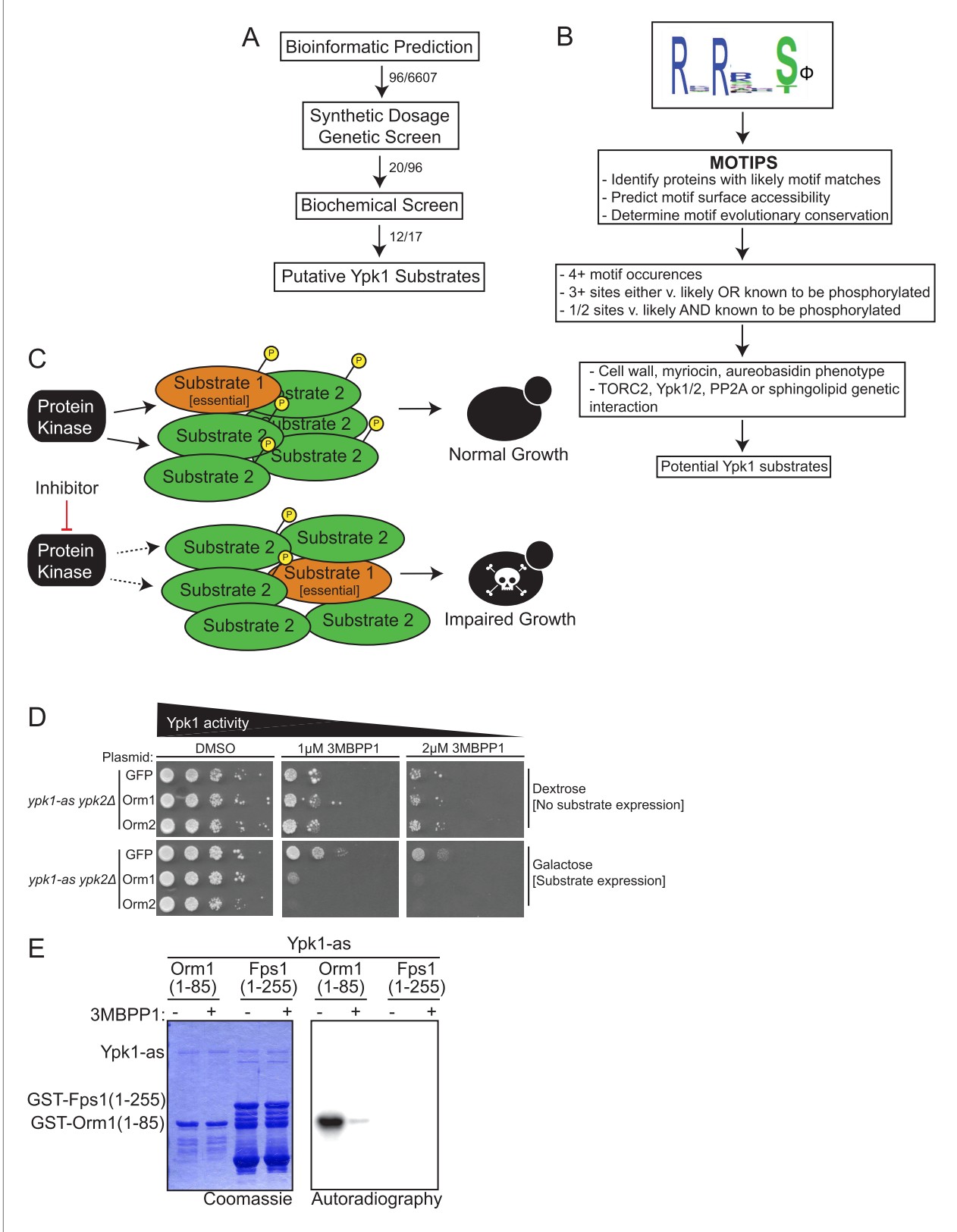

**Figure 1**. A three-part screen to identify likely Ypk1 substrates. (**A**) The three-part screening strategy to identify Ypk1 substrates is shown schematically as a flow chart. Numbers indicate the number of hits/considered genes at each step in the screen. (**B**) The bioinformatic approach towards identifying Ypk1 substrates is schematized as a flowchart with each filter as a box. Genes were first filtered by MOTIPS on the basis of having likely phosphorylatable Ypk1

*Figure 1. Continued on next page*

*Figure 1. Continued*

motifs. Subsequently, substrates were filtered by having many Ypk1 motifs or having a Ypk1 site known to be phosphorylated in published data sets. Lastly, genes were filtered by requiring the gene to have a published chemical sensitivity like Ypk1 does, or a published interaction with Ypk1, Ypk1 regulators (TORC2 or PP2A) or sphingolipid biosynthetic machinery. (**C**) A possible explanation for Ypk1 synthetic dosage lethality interactions is shown. Normally, the cell has enough kinase activity to buffer overexpression of a substrate (Substrate 2), so that essential substrates are regulated and normal growth is unperturbed. However, concurrent decrease in kinase activity coupled with substrate overexpression causes loss of regulation of essential substrate(s) (Substrate 1) leading to observable growth defects. (**D**) *ypk1-as ypk2Δ* (yAM135–A) cells were transformed with P*GAL1*-GFP (negative control), P*GAL1*-Orm1 or P*GAL1*-Orm2 (known Ypk1 substrates, positive SDL controls) plasmids. Overnight cultures were then serially diluted onto either dextrose (to repress substrate overexpression) or galatose (to induce substrate overexpression) containing media with increasing concentrations of the Ypk1-as inhibitor 3-MB-PP1. (**E**) GST-Orm1(1–85) (pFR203) and GST-Fps1(1–255) (pBT6) were purified from *E. coli* and incubated with [γ-$^{32}$P]ATP and Ypk1-as, purified from *S. cerevisiae*, in the absence or presence of 3-MB-PP1. The products were then resolved by SDS/PAGE and analyzed as described in 'Materials and methods'.

matches to the search logo, as predicted by MOTIPS. However, to avoid disregarding potential Ypk1 substrates with a single predicted match to the sequence logo, we also considered MOTIPS hits wherein there was existing evidence in the PhosphoGRID database (*Sadowski et al., 2013*) indicating that a predicted site is phosphorylated in vivo. To narrow down the list of potential substrates further, we chose to pursue those gene products containing matches to the sequence logo for which there was existing information in the literature suggesting a phenotypic relationship to known Ypk1-dependent processes: (a) a loss-of-function mutation in the candidate gene exhibits elevated sensitivity to agents (aureobasidin A, caspofungin, and/or myriocin) toward which a *ypk1Δ* mutant is also sensitive (*Hillenmeyer et al., 2008*); (b) the candidate gene product is reported to be involved in a genetic or biochemical interaction with Ypk1, as curated in YeastMine (*Balakrishnan et al., 2012*); (c) the candidate gene product is connected in some way to known Ypk1 regulators (*e.g.*, TORC2, PP2A); and/or, (d) the candidate gene product is involved in a known Ypk1-regulated process (*e.g.*, sphingolipid metabolism) (for further details, see 'Materials and methods'). Reassuringly, our approach identified three known Ypk1 substrates (Fpk1, Orm1 and Orm2); absence of Fpk2 and Gpd1 from the list generated solely by the MOTIPS search criteria arose from the fact that these substrates contain only a single predicted site that is not presently recorded in PhosphoGRID. For this reason, we also restored for consideration additional gene products that contain a single match to the consensus sequence logo that YeastMine indicated are involved in processes in which Ypk1 is implicated. The resulting candidates, grouped via cellular process on the basis of current GO Slim terminology (http://www.geneontology.org/GO.slims.shtml), are cataloged in *Table 1*, and represent fewer than 100 gene products out of the approximately 6000 protein-coding genes in the yeast genome (*Lin et al., 2013*) [although the number of authentic open-reading-frames undergoes constant revision (http://www.yeastgenome.org/cache/genomeSnapshot.html)].

As the secondary filter for the candidates recognized bioinformatically, we developed an in vivo approach to identify those gene products that manifested an expected hallmark of protein-substrate interaction. We reasoned that under conditions where the level of activity of an essential kinase, like Ypk1, is near-limiting for normal growth, high-level over-expression of an authentic substrate might tie up the available pool of active enzyme and prevent efficient phosphorylation of other cellular substrates necessary for growth and/or viability (*Figure 1C*). This scheme is a novel variation on a genetic approach referred to as synthetic dosage lethality (SDL) (*Sopko et al., 2006*; *Sharifpoor et al., 2012*). To limit Ypk1 activity, we used *ypk1-as ypk2Δ* cells, which express from the *YPK1* locus an analog-sensitive allele, Ypk1(L424A) (*Roelants et al., 2011*; *Niles et al., 2012*), and titrated down its activity by addition of a low concentration of an efficacious inhibitor,1-(*tert*-butyl)-3-(3-methylbenzyl)-1H-pyrazolo[3,4-d]pyrimidin-4-amine (3-MB-PP1) (*Burkard et al., 2007*) that has no effect on wild-type cells (see *Figure 2C*). To achieve high-level over-expression, each bioinformatic hit was expressed from the galactose-inducible *GAL1* promoter on a *CEN* plasmid. As proof of concept, we used two known Ypk1 substrates, Orm1 and Orm2, as positive controls and GFP, which is not a Ypk1 substrate (data not shown), as a negative control. In the absence of limiting the activity of Ypk1(L424A) with inhibitor, overexpression of neither Orm proteins nor GFP was deleterious to cell growth (*Figure 1D*, *left panels, compare lower to upper*). However, in the presence of a low dose of 3-MB-PP1, overexpression of Orm1 and Orm2 on galactose medium prevented cell growth, whereas overexpression of GFP did not (*Figure 1D*, *middle and right panels, compare lower to upper*). We were able to test the majority of

**Table 1.** Known Ypk1 substrates and potential substrates predicted by MOTIPS listed under GO Slim terms*

| Gene | MOTIPS sites | Chemical Sensitivity/YeastMine Interaction(s) | SDL score | Ypk1 dosage rescue | *In vitro* substrate |
|---|---|---|---|---|---|
| Known Ypk1 Substrates | | | | | |
| GPD1/YDL022W* | 24[P] | YeastMine | +++ | N/A | + |
| FPK1/YNR047W | 37, 200, 244, 436, 481 | YeastMine | + | N/A | + |
| FPK1(D621A) [Kinase-dead mutant] | 37, 200, 244, 436, 481 | YeastMine | ++ | N/A | + |
| ORM1/YGR038W | 52, 53 | YeastMine | +++ | N/A | + |
| ORM2/YLR350W | 47, 48 | YeastMine | ++++ | N/A | + |
| Cytoskeleton Organization | | | | | |
| AVO1/YOL078W | 552[P],597,1078 | YeastMine | – | N/A | N/A |
| AVO2/YMR068W | 273[P], 305, 407 | YeastMine | – | N/A | N/A |
| BEM2/YER155C | 83, 168, 1810, 1813 | Myr, YeastMine | – | N/A | N/A |
| BNI1/YNL271C | 119, 1138[P], 1533 | AbA, Casp, YeastMine | – | N/A | N/A |
| CDH1/YGL003C | 51, 195, 213[P] | AbA, YeastMine | TOXIC | – | N/A |
| ENT1/YDL161W | 160[P] | YeastMine | – | N/A | N/A |
| GIC2/YDR309C | 90, 312, 345[P] | Myr, AbA | – | N/A | N/A |
| LSB3/YFR024C-A | 262[P] | YeastMine | – | N/A | N/A |
| PAL1/YDR348C | 49[P], 391, 436 | AbA, Casp | ++++ | N/A | – |
| SLA1/YBL007C | 445, 447[P], 449[P], 477 | YeastMine | – | N/A | N/A |
| TSC11/YER093C | 19[P], 97, 188 | YeastMine | N/A | N/A | N/A |
| **YHR097C** | 58, 288[P], 294[P] | Myr | +++ | N/A | +/– |
| YSC84/YHR016C | 274, 374[P] | Myr, YeastMine | – | N/A | N/A |
| Biological Process Unknown | | | | | |
| COM2/YER130C | 251, 370, 380 | Myr, YeastMine | – | N/A | N/A |
| ECM3/YOR092W* | 312, 350 | Myr, AbA, YeastMine | – | N/A | N/A |
| ICS2/YBR157C | 14, 95, 136, 172[P] | Myr | – | N/A | N/A |
| JIP4/YDR475C | 348, 352, 592, 649 | Myr, AbA | N/A | N/A | N/A |
| KKQ8/YKL168C | 83, 113, 144, 146, 212, 293 | YeastMine | – | N/A | N/A |
| RTS3/YGR161C* | 30, 238[P] | YeastMine | – | N/A | N/A |
| SEG1/YMR086W | 56, 118[P], 217, 634, 752[P] | Myr | + | N/A | N/A |
| YDR186C | 334[P], 540[P], 542[P], 620, 715[P] | YeastMine | – | N/A | N/A |
| YHR080C | 401, 513, 667 | YeastMine | – | N/A | N/A |
| **YNR014W** | 54, 115, 156, 197 | YeastMine | +++ | N/A | + |
| YPK3/YBR028C | 72, 73, 90[P] | Myr, Casp | – | N/A | N/A |
| Transcription from RNA Polymerase II Promoter | | | | | |
| EPL1/YFL024C | 24, 28, 61 | YeastMine | – | N/A | N/A |
| FKH1/YIL131C | 404 | Myr, YeastMine | TOXIC | – | – |
| GAL11/YOL051W | 1003[P] | Myr, YeastMine | – | N/A | N/A |
| HCM1/YCR065W* | 80 | AbA, YeastMine | – | N/A | N/A |
| HOT1/YMR172W* | 387, 520, 586 | Myr | – | N/A | N/A |
| RLM1/YPL089C* | 20 | Myr | TOXIC | – | N/A |

*Table 1. Continued on next page*

*Table 1. Continued*

| Gene | MOTIPS sites | Chemical Sensitivity/YeastMine Interaction(s) | SDL score | Ypk1 dosage rescue | *In vitro* substrate |
|---|---|---|---|---|---|
| **SMP1/YBR182C*** | 20, 107 | YeastMine | TOXIC | + | + |
| SSN2/YDR443C | 608[P] | Myr | − | N/A | N/A |
| YHP1/YDR451C* | 180, 182 | Myr | − | N/A | N/A |
| Mitotic Cell Cycle | | | | | |
| BCK2/YER167W | 12, 38, 373, 430 | YeastMine | TOXIC | − | N/A |
| ESC2/YDR363W | 114, 143, 145 | YeastMine | − | N/A | N/A |
| PTK2/YJR059W | 59[P], 82, 91, 171, 275 | Myr, YeastMine | + | N/A | N/A |
| SET3/YKR029C | 236, 405, 428 | YeastMine | − | N/A | N/A |
| SWI4/YER111C | 816[P] | Myr, YeastMine | − | N/A | N/A |
| VHS2/YIL135C | 316, 318, 325[P] | Myr, Casp | − | N/A | N/A |
| ZDS1/YMR273C* | 78, 370 | AbA, YeastMine | − | N/A | N/A |
| ZDS2/YML109W | 183, 267, 345 | YeastMine | − | N/A | N/A |
| Protein Phosphorylation | | | | | |
| HAL5/YJL165C | 17[P], 217[P], 233 | YeastMine | − | N/A | N/A |
| KIN1/YDR122W | 652, 791[P], 879, 986[P] | YeastMine | + | N/A | N/A |
| KIN2/YLR096W | 665[P], 818, 1020 | Myr | − | N/A | N/A |
| KSP1/YHR082C | 594, 827[P], 884[P] | Myr, AbA, YeastMine | TOXIC | − | N/A |
| NPR1/YNL183C | 125[P], 255[P], 257[P], 317[P] | YeastMine | ++++ | N/A | − |
| SKY1/YMR216C | 383[P] | Myr, YeastMine | − | N/A | N/A |
| YAK1/YJL141C | 128[P], 206, 240 | Myr, Casp, YeastMine | − | N/A | N/A |
| YPL150W | 371[P], 452, 890 | YeastMine | − | N/A | N/A |
| Lipid Metabolic Process | | | | | |
| ADR1/YDR216W | 180, 230[P], 756 | Myr, AbA | − | N/A | N/A |
| CDC1/YDR182W* | 9 | N/A | − | N/A | + |
| CKI1/YLR133W | 14[P], 25[P], 30[P] | Myr, AbA, YeastMine | − | N/A | N/A |
| **GPT2/YKR067W** | 27, 652 | Myr | +++ | N/A | + |
| **LAC1/YKL008C*** | 23, 24 | Myr, YeastMine | +++ | N/A | + |
| **LAG1/YHL003C** | 24[P] | Myr, YeastMine | +++ | N/A | + |
| LCB3/YJL134W | 16[P] | Myr, YeastMine | − | N/A | + |
| Cellular Ion Homeostasis and Transport | | | | | |
| AVT3/YKL146W | 55, 59[P], 172, 174 | Myr, AbA, YeastMine | − | N/A | N/A |
| CCH1/YGR217W† | 146, 148, 347 | YeastMine | − | N/A | + |
| **FPS1/YLL043W** | 147, 181, 185, 570[P] | Myr, YeastMine | +++++ | N/A | + |
| MNR2/YKL064W | 165, 620, 621, 826 | AbA | − | N/A | N/A |
| NHA1/YLR138W* | 544, 830 | Myr, YeastMine | − | N/A | N/A |
| PPZ1/YML016C | 122, 203, 250[P] | Myr, YeastMine | TOXIC | − | N/A |
| Translation | | | | | |
| DED1/YOR204W | 84, 576[P] | YeastMine | − | N/A | N/A |
| HCR1/YLR192C* | 223 | Myr, AbA, YeastMine | − | N/A | N/A |
| HEF3/YNL014W* | 898 | Myr, AbA, YeastMine | − | N/A | N/A |
| RPL3/YOR063W | 24[P], 337 | Myr, AbA, YeastMine | − | N/A | N/A |
| SUI2/YJR007W* | 58 | Myr | − | N/A | N/A |
| TEF1/YPR080W* | 72[P] | Myr | − | N/A | N/A |

*Table 1. Continued on next page*

*Table 1. Continued*

| Gene | MOTIPS sites | Chemical Sensitivity/YeastMine Interaction(s) | SDL score | Ypk1 dosage rescue | *In vitro* substrate |
|---|---|---|---|---|---|
| **Cell Wall Organization or Biogenesis** | | | | | |
| BPH1/YCR032W | 1334, 1336, 1963 | Casp | – | N/A | N/A |
| CSR2/YPR030W | 61, 103, 525, 987 | Myr | – | N/A | N/A |
| ROM2/YLR371W | 76[P], 193[P], 396 | YeastMine | – | N/A | N/A |
| SSD1/YDR293C | 164[P], 482[P], 503 | Myr, AbA, YeastMine | TOXIC | – | N/A |
| **Golgi Vesicle Transport** | | | | | |
| BRE5/YNR051C | 398[P] | Myr, YeastMine | +++ | N/A | – |
| EXO84/YBR102C | 76, 313, 494, 554 | YeastMine | – | N/A | N/A |
| **MUK1/YPL070W** | 173, 184[P], 185[P] | Myr | +++ | N/A | +/– |
| RGP1/YDR137W | 220, 364[P], 450, 452 | YeastMine | – | N/A | N/A |
| **Signaling** | | | | | |
| IRA2/YOL081W | 882, 884, 1578, 1745, 3069 | YeastMine | N/A | N/A | N/A |
| GIS3/YLR094C* | 249, 333 | Myr, AbA | – | N/A | N/A |
| MDS3/YGL197W | 757, 824, 842, 851, 1204 | Myr, AbA, YeastMine | +++ | N/A | – |
| SYT1/YPR095C | 277[P], 410, 728 | YeastMine | N/A | N/A | N/A |
| **DNA Replication** | | | | | |
| CDC13/YDL220C | 314, 333[P] | YeastMine | TOXIC | – | N/A |
| CTI6/YPL181W | 155, 216[P] | Myr, YeastMine | – | N/A | N/A |
| RIM4/YHL024W | 93, 429, 525, 607 | YeastMine | – | N/A | N/A |
| **Endocytosis** | | | | | |
| ALY2/YJL084C | 166[P], 201, 225, 803 | Myr | – | N/A | N/A |
| **ROD1/YOR018W** | 563, 617, 807, 823 | Myr | +++ | N/A | +/– |
| ROG3/YFR022W | 425, 584, 718 | YeastMine | – | N/A | N/A |
| **Other** | | | | | |
| FRT1/YOR324C | 167, 201, 203, 228[P], 385 | Myr, YeastMine | – | N/A | N/A |
| HER1/YOR227W | 28[P], 102[P], 157[P] | Myr, AbA | TOXIC | – | + |
| **YSP2/YDR326C** | 326, 518, 1237 | Myr, YeastMine | +++ | N/A | + |
| **RNA Catabolic Process** | | | | | |
| JSN1/YJR091C | 174, 275[P], 600 | YeastMine | – | N/A | N/A |
| PUF2/YPR042C | 55, 143, 246, 902 | Myr | N/A | N/A | N/A |
| **Cytokinesis** | | | | | |
| CYK3/YDL117W | 159, 207[P], 746 | AbA, YeastMine | +++ | N/A | – |
| **Chromosome Segregation** | | | | | |
| DSN1/YIR010W | 240, 250[P] | YeastMine | – | N/A | N/A |
| **Peroxisome Organization** | | | | | |
| **PEX31/YGR004W** | 432[P] | YeastMine | ++ | N/A | + |
| **Pseudohyphal Growth** | | | | | |
| PAM1/YDR251W | 471, 553[P], 625 | Myr, AbA, Casp, YeastMine | – | N/A | N/A |
| **Response to Starvation** | | | | | |
| **ATG21/YPL100W** | 191, 237[P] | Myr | +++ | N/A | + |

*Genes that are not bioinformatically predicted Ypk1 substrates, but contain Ypk1 motifs and were included in this study are marked with an asterisk. SDL assay results are listed for each bioinformatically predicted Ypk1 substrate. The scoring system reports growth phenotypes of the *ypk1-as ypk2Δ* strain transformed with the indicated

*Table 1. Continued on next page*

Muir *et al*. eLife 2014;3:e03779. DOI: 10.7554/eLife.03779

*Table 1. Continued*

P$_{GAL1}$-SUBSTRATE plasmid upon overexpression on galactose with varying levels of 3-MB-PP1-imposed Ypk1-as inhibition. A growth phenotype is defined as at least 1 serial dilution spot less growth than YCpLG-GFP control at the given 3-MB-PP1 concentration. A strong growth phenotype is defined as no growth at the given 3-MB-PP1 concentration. (+++++) indicates a growth phenotype with no 3-MB-PP1. (++++) is a strong growth phenotype on 1 μM 3-MB-PP1. (+++) indicates a growth phenotype on 1 μM 3-MB-PP1. (++) is defined as no phenotype on 1 μM 3-MB-PP1, but a strong growth phenotype on 2 μM 3-MB-PP1. (+) indicates no phenotype on 1 μM 3-MB-PP1, but a detectable growth phenotype on 2 μM 3-MB-PP1. (−) indicates no growth phenotype at any concentration of 3-MB-PP1 tested. TOXIC indicates overexpression of the putative substrate on galactose-containing medium was deleterious to growth even in the wild-type control strain (BY4741). These toxic genes were then tested for Ypk1 dosage rescue (for details, see 'Materials and methods'); here, (+) indicates that Ypk1 overexpression could at least partially rescue the overexpression toxicity of the indicated gene and (−) indicates that Ypk1 overexpression could not rescue the overexpression toxicity. Lastly, the results of testing the indicated purified predicted Ypk1 target as a substrtate in the in vitro protein kinase assay with purified Ypk1-as; here, (+) indicates that Ypk1-as- dependent (3-MB-PP1 inhibitable) incorporation was detectable for the substrate at a level comparable to incorporation into the positive control [the known Ypk1 substrate, GST-Orm1(1–85)]; (+/−) indicates that readily detectable Ypk1-as-dependent incorporation was found, but at a level lower than that seen for an equivalent amount of GST-Orm1(1–85) protein. (N/A) indicates that the indicated gene product was not tested in the indicated assay.
†The SDL assay was performed with a plasmid constitutively overexpressing *CCH1* under the *TDH3* promoter [pBCT-CCH1H, (**Iida et al., 2007**)], as our efforts to generate a P$_{GAL1}$-*CCH1* vector failed.

the candidates (90/96) that arose in the bioinformatic search in this same fashion [however, 10/90 caused toxicity upon over-expression even in wild-type cells and, hence, could not be scored].

Those candidates that, like Orm1 and Orm2, exhibited toxicity only on galactose medium and only when Ypk1(L424A) activity was limited in the presence of 3-MB-PP1, but not when inhibitor was absent, were designated SDL hits (**Table 1**, *column 4*). Moreover, use of a series of 3-MB-PP1 concentrations allowed for quantification of the strength of the SDL effect (from + to ++++). In one case (Fps1), a marked SDL effect was observed upon overexpression in the *ypk1-as ypk2Δ* cells in the absence of chemical inhibition; we considered this a valid SDL hit because *GAL* promoter-driven over-expression of Fps1 was not growth inhibitory in wild-type (*YPK1+ YPK2+*) cells. Thus, as summarized in **Table 1** (*column 4*), a significant fraction (20/90) of the candidates identified bioinformatically that were tested in this fashion, but far from all, displayed an SDL phenotype. In this regard, it is important to note that all known Ypk1 substrates tested (Gpd1, Fpk1, Orm1 and Orm2) displayed an SDL phenotype, whereas nearly 80% of the bioinformatic hits, like GFP, did not. Consistent with the view that the SDL phenotype could arise from the over-expressed target serving as a decoy substrate that titers a limited pool of active Ypk1 away from acting on its essential substrates, we observed that over-expressed catalytically-inactive Fpk1 caused an SDL phenotype equivalent to or stronger than wild-type Fpk1 (**Table 1**). If such SDL phenotypes reflect occlusion of a limited pool of enzyme by over-expressed substrate, then, conversely, co-overexpression of Ypk1 or even of a kinase-dead allele Ypk1(K376A) (driven from the *MET25* promoter) might rescue the toxicity. Indeed, the deleterious effect of Smp1 over-expression was rescued by co-overexpression of either Ypk1 or Ypk1(K376A) (**Table 1**), suggesting that the SDL phenotype of over-expressed Smp1 also arises from titration of a limited amount of Ypk1 away from essential substrates.

Lastly, to determine whether the gene products that displayed an SDL phenotype are indeed substrates for Ypk1, we incubated those (17/20) that we were able to successfully express and purify as recombinant proteins or protein fragments (as GST fusions) from *Escherichia coli* with [γ-$^{32}$P]ATP and Ypk1(L424A), which was highly purified from yeast cells as described in 'Materials and methods'. We chose to use the analog-sensitive allele, even though it is only about 50% as active as wild-type Ypk1 (**Roelants et al., 2011**), because ablation of activity in the presence of 3-MB-PP1 allowed us to confirm that any $^{32}$P incorporation into substrate observed was due to phosphorylation by Ypk1(L424A) itself (and not due to some other protein kinase that might be present in the preparation). All known in vivo substrates of Ypk1 (Fpk1, Fpk2, Gpd1, Orm1 and Orm2) display robust incorporation (as judged by autoradiography) in this in vitro assay (**Roelants et al., 2010, 2011**; **Lee et al., 2012b**). Therefore, in testing each candidate, an appropriate positive control, Orm1(1–85) was included (**Figure 1E**), which also allowed comparison between independent assays. Gratifyingly, 12/17 (70%) of the SDL hits tested in this manner displayed readily detectable and Ypk1-specific phosphorylation (**Table 1**, *right column*).

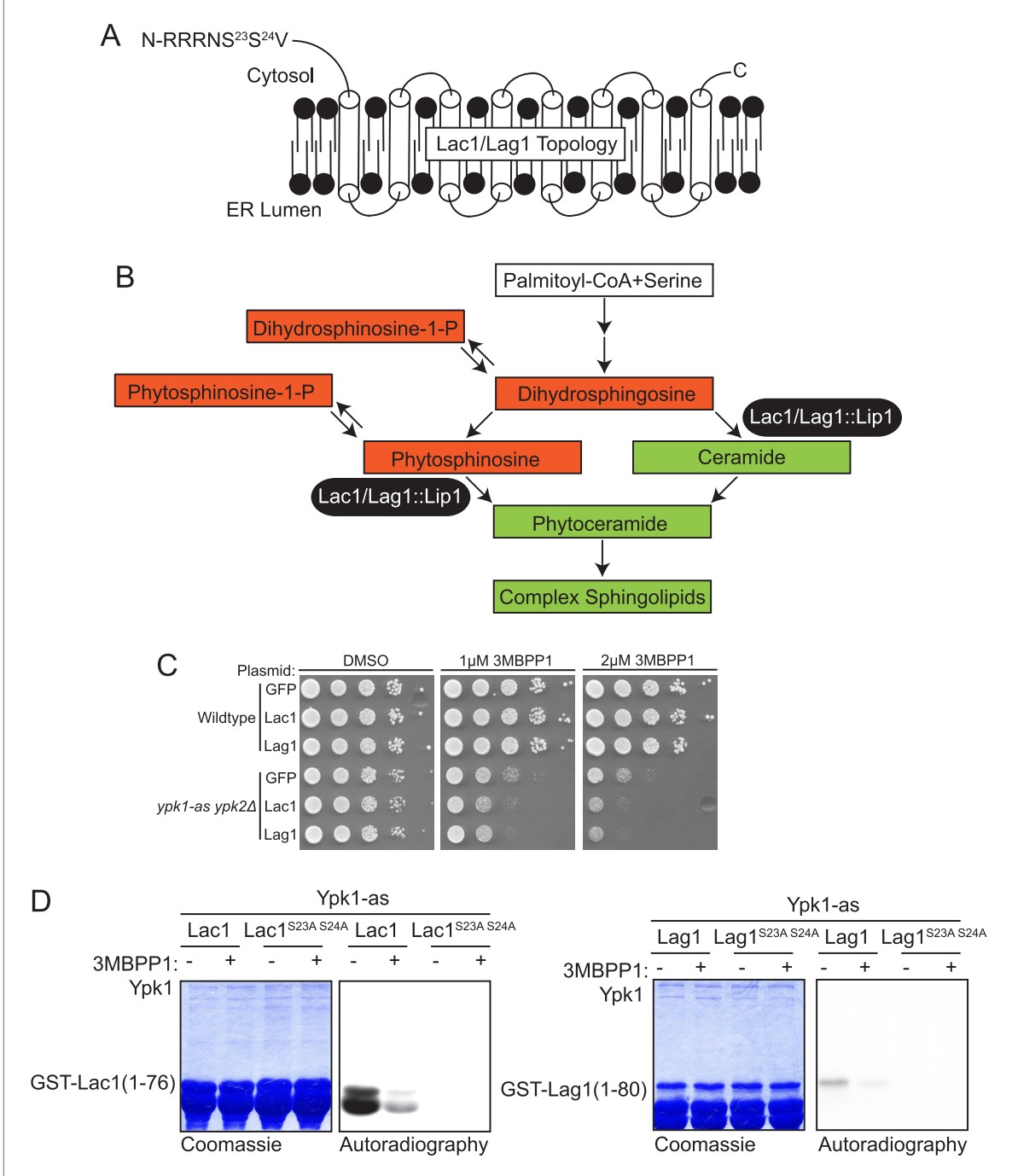

**Figure 2**. Lac1 and Lag1, subunits of ceramide synthase, were identified by the screen. (**A**) A diagram of the membrane topology of Lac1 and Lag1 derived from (***Kageyama-Yahara and Riezman, 2006***); Lac1 and Lag1 are experimentally determined to have eight transmembrane domains. The N terminus of these proteins is cytosolic and therefore accessible for Ypk1 phosphorylation. (**B**) Diagram of yeast de novo sphingolipid biosynthesis derived from (***Dickson, 2008***). Metabolites appear as boxes and enzymes as ovals. Metabolites in green are those directly produced or derived from ceramide synthase while those in red are alternative products at the ceramide synthase branch point. (**C**) SDL results for Lac1 and Lag1. *ypk1-as ypk2Δ* (yAM135–A) or wildtype (BY4741) cells were transformed with P$_{GAL1}$-GFP (negative control), P$_{GAL1}$-Lac1 or P$_{GAL1}$-Lag1 plasmids. The SDL assay was then performed as described in 'Materials and methods'. (**D**) GST-Lac1(1–76) (pAX131), GST-Lag1(1–80)(pFR29), GST-Lac1(1–76)(S23A S24A)(pAX132) and GST-Lag1(1–80)(S23A S24A)(pAX133) were purified from *E. coli* and Ypk1-as kinase assays were performed as in *Figure 1D*.

Moreover, the results of such assays clearly demonstrated that Ypk1 is not a 'promiscuous' enzyme. For example, Fps1 has three possible N-terminal Ypk1 sites (**R**PR**G**Q**T**[147]T, **RRR**SR**S**[181]R and **R**SRAT**S**[185]N) and one C-terminal site. The C-terminal site is a Ypk1 target (data not shown); however, none of the N-terminal motifs serves as a Ypk1 phospho-acceptor site (*Figure 1E*), most likely because each lacks a hydrophobic residue at +1.

Thus, we considered it very likely that the dozen candidates identified bioinformatically that also displayed an SDL phenotype and served as Ypk1 substrates in vitro (highlighted in bold in *Table 1*) would be functionally important Ypk1 substrates in vivo. To validate this conclusion and confirm that these candidates are indeed physiologically relevant Ypk1 targets, we chose to characterize Lac1 and Lag1, two of the dozen candidates (*Table 1*), because they are the catalytic subunits of the ceramide synthase complex and might further our understanding about how sphingolipid production is regulated by the TORC2-Ypk1 signaling axis.

## Ceramide synthase components Lac1 and Lag1 are identified as Ypk1 substrates

Lac1 (418 residues) and Lag1 (411 residues) are apparent paralogs at the primary sequence level (69% identity, 77% similarity) (*Byrne and Wolfe, 2005*) and are polytopic integral proteins in the ER membrane with the predicted Ypk1 site in each protein residing near its N-terminus (*Figure 2A*). It has been shown that the N-termini of these proteins are exposed to the cytosol (*Kageyama-Yahara and Riezman, 2006*). Along with a small accessory subunit Lip1 (150 residues), Lac1 and Lag1 are demonstrated constituents of the ceramide synthase complex (*Schorling et al., 2001*; *Vallée and Riezman, 2005*), which catalyzes N-acylation of the free amino group on the long-chain base (LCB), mainly phytosphingosine in yeast, using fatty acyl-CoA as the acyl donor, thereby forming phytoceramide (*Figure 2B*). Genetically, Lac1 and Lag1 appear to play overlapping functional roles; *lac1Δ* or *lag1Δ* single mutants are viable, whereas a *lac1Δ lag1Δ* double mutant is reportedly either inviable (*Jiang et al., 1998*) or extremely slow growing (*Barz and Walter, 1999*; *Schorling et al., 2001*; *Vallée and Riezman, 2005*). The ceramide synthase reaction lies at an important branch point in the sphingolipid metabolic network (*Figure 2B*) because de novo synthesis of ceramides both consumes LCBs and prevents conversion of LCBs to their 1-phosphorylated derivatives (LCBPs). Thus, the rate of ceramide synthesis is tightly coupled to the levels of both LCBs and LCBPs (*Kobayashi and Nagiec, 2003*); and, moreover, the balance between ceramides and total LCBs and LCBPs affects growth rate in both fungi (*Kobayashi and Nagiec, 2003*) and mammalian cells (*Spiegel and Milstien, 2003*). By virtue of their position in the pathway, Lac1 and Lag1 are situated to be important regulators of this balance.

Among the bioinformatically predicted substrates, both Lac1 and Lag1 displayed a readily detectable SDL phenotype (*Figure 2C*) and both GST-Lac1(1–76) and GST-Lag1(1–80) served as in vitro substrates for Ypk1, albeit with the phosphorylation of Lac1 being reproducibly much more robust than Lag1 (*Figure 2D*). Site-directed mutagenesis confirmed that the Ypk1-mediated phosphorylation of both substrates occurred exclusively at the predicted phospho-acceptor site(s), specifically Ser23 and Ser24 in both proteins (*Figure 2D*).

## Lac1 and Lag1 are phosphorylated by Ypk1 in vivo

To determine whether both Lac1 and Lag1 are phosphorylated in vivo in a Ypk1-dependent manner and at their Ypk1 consensus sites, integrated 3xHA- or 3xFLAG-tagged versions of each protein and its corresponding S23A S24A mutant were expressed in yeast and extracts of the cells were analyzed by phosphate-affinity SDS-PAGE (Phos-tag gels) (*Kinoshita et al., 2009*). In this separation technique, the more highly phosphorylated the protein, the more its migration is retarded. Both Lac1 (*Figure 3A*, *left*) and Lag1 (*Figure 3A*, *right*) migrated as two species, and the slower of the two could be attributed to phosphorylation because it was eliminated if the sample was pre-treated with calf intestinal phosphatase. This slower mobility species represented phosphorylation at S23 and S24 because the band was also eliminated in Lac1(S23A S24A) and Lag1(S23A S24A) mutants (*Figure 3A*). We noted that phosphatase treatment, even of the Lac1(S23A S24A) and Lag1(S23A S24A) mutants, resulted in appearance of a third, even faster migrating species, presumably due to removal of a phosphorylation(s) elsewhere in these proteins, consistent with indirect evidence that Lac1 and Lag1 might be subject to casein kinase II (yeast Cka2)-dependent modification (*Kobayashi and Nagiec, 2003*).

In agreement with their relative efficacies as in vitro substrates (*Figure 2D*), we found that, reproducibly, the majority of Lac1 was present in the cell as the slower mobility isoform, whereas the opposite

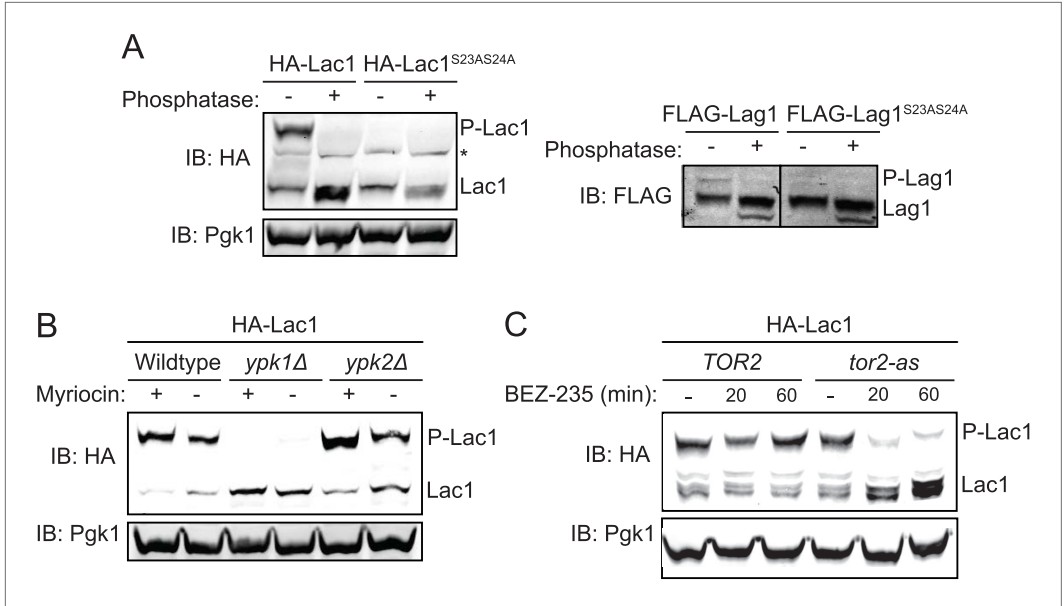

**Figure 3**. Ypk1 phosphorylates Lac1 and Lag1 at S23 and S24 in vivo. (**A**) 3xHA-Lac1 (yAM165–A) 3xHA-Lac1(S23A S24A) (yAM166–A), 3xFLAG-Lag1 (yAM159–A) and 3xFLAG-Lag1(S23A S24A) (yAM163–A) strains were grown to mid-exponential phase in YPD. Cells were then harvested and whole-cell extracts were prepared. Extracts were split and one fraction was then treated with calf intestinal phosphatase. The extracts resolved by Phos-tag SDS-PAGE and immunoblotted with anti-HA, -FLAG or -Pgk1 (loading control) antibodies. P-Lac1 and P-Lag1 indicate the band corresponding to S23 S24 phosphorylation. * indicates a non-specific band that appears in HA blots of yeast whole cell extracts. (**B**) Wildtype (BY4741), *ypk1Δ* (JTY6142) and *ypk2Δ* (yAM120–A) strains were transformed with a plasmid centromeric plasmid encoding 3xHA-Lac1 expressed under control of its endogenous promoter (pAX136). Cells were grown to mid-exponential phase and then treated with a sublethal dose of myriocin (0.625 μM) or methanol (vehicle) for 2 hr. Cell extracts were then prepared, resolved by Phos-tag SDS-PAGE and blotted as above in (**A**). (**C**) *TOR2* (yKL04) or *tor2-as* (yKL05) strains were transformed with a 3xHA-Lac1 expressing plasmid (pAX136). Cells were grown to mid-exponential phase and treated with 1 μM BEZ-235 for the indicated times. Cell extracts were then prepared, resolved by Phostag SDS-PAGE and blotted as above in (**A**).

was true for Lag1 (*Figure 3A*). Because the behavior of Lac1 gave us greater sensitivity of detection, and for the sake of conciseness, some of our subsequent findings are illustrated with data for Lac1 only. However, all experiments were repeated with both proteins with virtually identical results and conclusions.

To further confirm that Ypk1 is the protein kinase responsible for phosphorylation at Ser23 Ser24 in vivo, plasmids encoding 3xHA-tagged Lac1 and Lag1 were introduced by DNA-mediated transformation into wild-type, *ypk1Δ*, and *ypk2Δ* strains. Cultures of the resulting cells were grown in the absence or presence of a sub-lethal dose of myriocin, a condition which several previous studies have shown activates TORC2- and Ypk1-mediated signaling (*Roelants et al., 2011*; *Berchtold et al., 2012*; *Sun et al., 2012*), and the resulting extracts were analyzed on Phos-tag gels. As observed previously for two other *bona fide* substrates, Orm1 and Orm2 (*Roelants et al., 2011*), absence of Ypk1 totally abrogated the appearance of phosphorylated Lac1 (*Figure 3B*) and phosphorylated Lag1 (data not shown), whereas elimination of Ypk2 had no effect. Thus, Ypk1 is the paralog solely responsible for the observed in vivo phosphorylation of Lac1 and Lag1 at Ser23 and Ser24. Furthermore, under conditions that stimulate TORC2 activity, nearly all of the Lac1 (*Figure 3B*) and much more of the Lag1 (data not shown) were converted to the phosphorylated isoform indicating that TORC2 activation is relayed to ceramide synthase via Ypk1. To further confirm that TORC2 function is essential for Ypk1-mediated phosphorylation of ceramide synthase, Lac1 phosphorylation was monitored in *TOR2* cells and in a *tor2-as* mutant after addition of a specific *tor2-as* inhibitor, BEZ-235 (*Kliegman et al., 2013*). TORC2 inhibition markedly reduced Lac1 phosphorylation within 20 min in *tor2-as* cells, but not in the otherwise isogenic control cells (*Figure 3C*), confirming that TORC2 activity is necessary for Ypk1-mediated

ceramide synthase phosphorylation. Thus, our screening approach was successful in revealing two, previously uncharacterized, cellular targets of the TORC2-Ypk1 signaling axis.

## Increased Lac1 and Lag1 phosphorylation is required for cell survival under stress

The fact that impeding LCB production with a sub-lethal dose of the SPT inhibitor myriocin stimulated Ypk1-mediated Lac1 and Lag1 phosphorylation suggested that this modification is important for their physiological function. As one means to confirm that reduction in sphingolipid biosynthesis capacity results in up-regulation of Lac1 and Lag1 phosphorylation, we subjected the pathway to blockade near its end by treating the cells expressing integrated 3xHA-tagged Lac1 or Lag1 with aureobasidin A (*Heidler and Radding, 1995*), an antibiotic that prevents formation of complex sphingolipids in yeast by inhibiting Aur1 (phosphatidylinositol:ceramide phosphoinositol transferase) (*Nagiec et al., 1997*). As observed for treatment with myriocin, the amount of phosphorylated Lac1 (*Figure 4A*) and Lag1 (data not shown) was markedly increased in response to treatment with aureobasidin A.

Another perturbation that has been shown to transiently up-regulate TORC2-Ypk1-mediated signaling is heat shock (*Sun et al., 2012*). Consistent with that response, it has been shown previously that heat shock leads to a transient increase in sphingolipid production and that sphingolipid production is important for heat shock survival (*Jenkins et al., 1997*; *Cowart et al., 2003*). Moreover, measurement of pathway intermediates and mathematical modeling also suggested that a sharp spike of increased ceramide synthase activity may occur after heat shock (*Chen et al., 2013*). Therefore, we subjected the same cells to heat shock and monitored Lac1 and Lag1 phosphorylation at various times thereafter. Within 5 min, the amount of phosphorylated Lac1 (*Figure 4B*) and Lag1 (data not shown) increased markedly, but was back to the resting level by 30 min.

If these changes in phosphorylation state at Ser23 and Ser24 in Lac1 and Lag1 are important for the metabolic adjustments that the cell needs to adapt appropriately, then preventing phosphorylation at these sites should impair cell survival. To test this prediction, we integrated as the sole source of ceramide synthase, mutant versions of Lac1 and Lag1 in which Ser23 and Ser24 were mutated to Ala and, hence, cannot be phosphorylated under any circumstances. As a control, we also generated integrated versions of Lac1 and Lag1 in which Ser23 and Ser24 were mutated to Glu, to mimic conversion of the entire population to the phosphorylated state, a response that we showed can be achieved for the wild-type protein (see, for example, *Figure 3B*). Indeed, we found that the cells co-expressing Lac1(S23A S24A) and Lag1(23A S24A) grew detectably less well when challenged with either myriocin or aureobasidin A than either otherwise isogenic wild-type cells or cells co-expressing Lac1(S23E S24E) and Lag1(S23E S24E) (*Figure 4C*). These growth differences could not be attributed to differences in the level of expression of these proteins, as immunoblot analysis demonstrated the wild-type and mutant ceramide synthase subunits were present in equivalent amounts under the conditions tested (*Figure 4D*). Thus, Ypk1-dependent phosphorylation of these sites in Lac1 and Lag1 is functionally important for cell survival in response to the stress of limiting sphingolipid biosynthesis. Furthermore, the fact that the Lac1(S23E S24E) Lag1(S23E S24E) strain phenocopied a *LAC1⁺ LAG1⁺* strain under conditions that promote phosphorylation of Lac1 and Lag1 indicates that these mutations generated effective phosphomimetic alleles.

## Calcineurin down-regulates Lac1 and Lag1 phosphorylation at Ser23 and Ser24

As a means to delineate what cellular phosphatase is responsible for counteracting the Ypk1-mediated phospho-regulation of Lac1 and Lag1, plasmid-borne 3xHA-tagged Lac1 was expressed in a collection of deletion strains lacking each of the non-essential protein phosphatase genes or their associated factors, and the phosphorylation state of Lac1 was assessed using Phos-tag gels. By this approach, we were unable to find any phosphatase-deficient mutant that exhibited a significant increase in the amount of phosphorylated Lac1 compared to control cells (data not shown). However, considerable genetic evidence indicates a strong connection between Ca²⁺ signaling and sphingolipid biosynthesis (*Beeler et al., 1998*). Moreover, it has been reported previously that TORC2-Ypk-dependent regulation of sphinglipid biosynthesis is antagonized by the action of the Ca²⁺/calmodulin-dependent protein phosphatase calcineurin (also known as phosphoprotein phosphatase 2B), although the level at which the phosphatase acted was unknown (*Aronova et al., 2008*). Hence, we conducted the converse experiment by stimulating the cells expressing 3xHA-tagged Lac1 acutely with 0.2 M CaCl₂, a

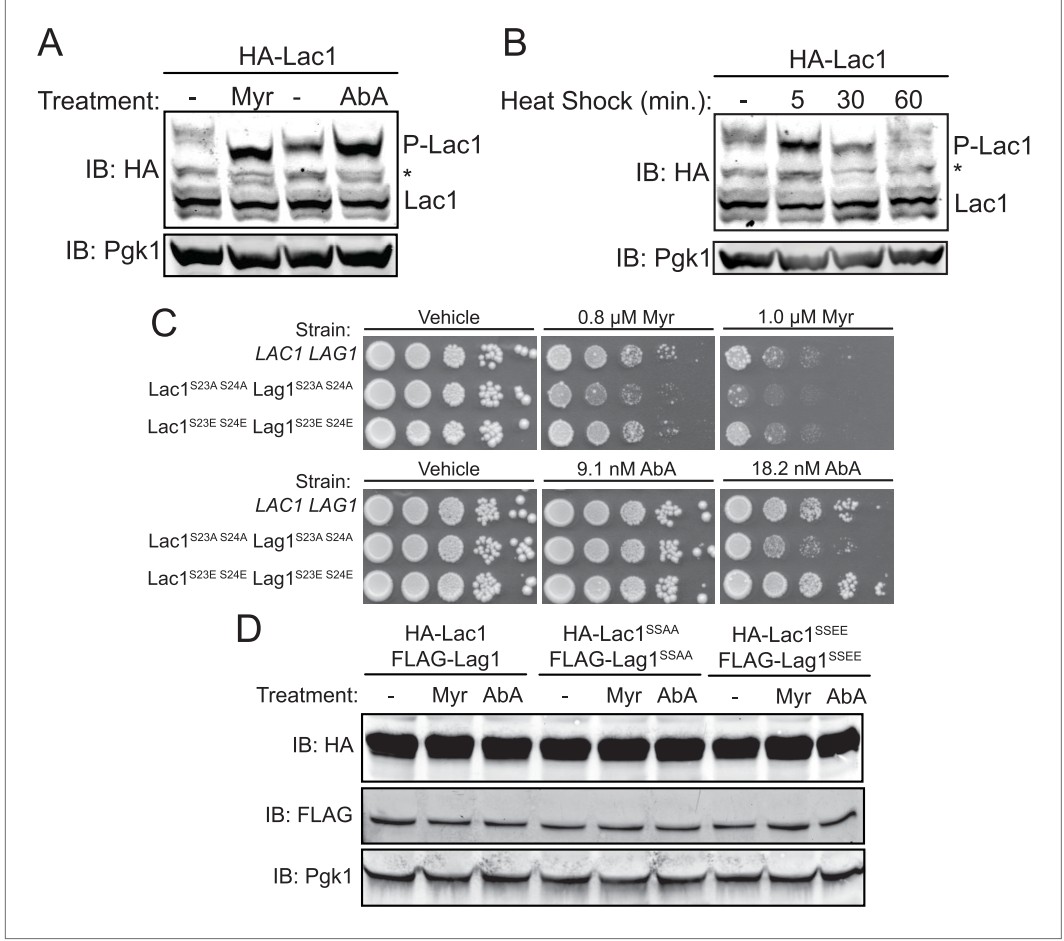

**Figure 4**. Enhanced Ypk1 phosphorylation of Lac1 and Lag1 under sphingolipid and heat stress. (**A**) 3xHA-Lac1 (yAM165–A) cells were grown to early exponential phase in YPD. Cultures were then treated with sublethal doses of myriocin (0.625 μM) or methanol (vehicle) or aureobasidin A (1.8 μM) or ethanol (vehicle) for 2 hr. (**B**) 3xHA-Lac1 (yAM165–A) cells were grown to exponential phase in YPD at 30°C. A sample of this culture was then harvested. The remaining culture was then moved to 42°C to initiate heat shock and samples were harvested at the indicated time points. Whole cell extracts were prepared from each sample, resolved by Phos-tag SDS-PAGE, and immunob-lotted as in *Figure 3*. P-Lac1 indicates the band corresponding to S23 S24 phosphorylation. * indicates a non-specific band that appears in HA blots of yeast whole cell extracts. (**C**) *LAC1 LAG1* (yAM205–A), Lac1SSAA Lag1SSAA (yAM207B) and Lac1SSEE Lag1SSEE (yAM210) were grown to exponential phase in YPD. Serial dilutions of each culture were made and spotted on YPD plates containing vehicle or the indicated concentration of myriocin or aureobasi-din A. Cells were allowed to grow for 3 days at 30°C prior to imaging. (**D**) 3xHA-Lac1::*HIS3* 3xFLAG-Lag1::*LEU2* (yAM168) 3xHA-Lac1(S23A S24A)::*HIS3* 3xFLAG-Lag1(S23A S24A)::*LEU2* (Lac1SSAA Lag1SSAA) (yAM184) and 3xHA-Lac1(S23E S24E)::*HIS3* 3xFLAG-Lag(S23E S24E)1::*LEU2* (Lac1SSEE Lag1SSEE) (yAM192–A) cells were grown to mid-exponential phase and then treated with 1.0 μM myriocin or 18.2 nM aureobasidin A for 2 hr. Whole cell extracts were prepared from each sample, resolved by SDS-PAGE, and immunoblotted as indicated.

condition known to robustly activate calcineurin in yeast (*Stathopoulos-Gerontides et al., 1999*). Within <10 min, we found total abrogation of phospho-Lac1 (*Figure 5A*, *left*) and total abrogation of phospho-Lag1 (data not shown) in wild-type cells, whereas in otherwise isogenic *cna1Δ cna2Δ* mutants (which lack both calcineurin catalytic subunit paralogs) a substantial portion of the phosphorylated species remained (*Figure 5A*, *right*).

These results suggested that calcineurin may directly reverse the phoshorylations introduced into Lac1 and Lag1 by Ypk1. However, it is also the case that two of the ancillary subunits associated with TORC2, Slm1 and Slm2, are demonstrated calcineurin-binding proteins (*Bultynck et al., 2006*; *Tabuchi et al., 2006*) and that calcineurin action appears to oppose TORC2-dependent signaling

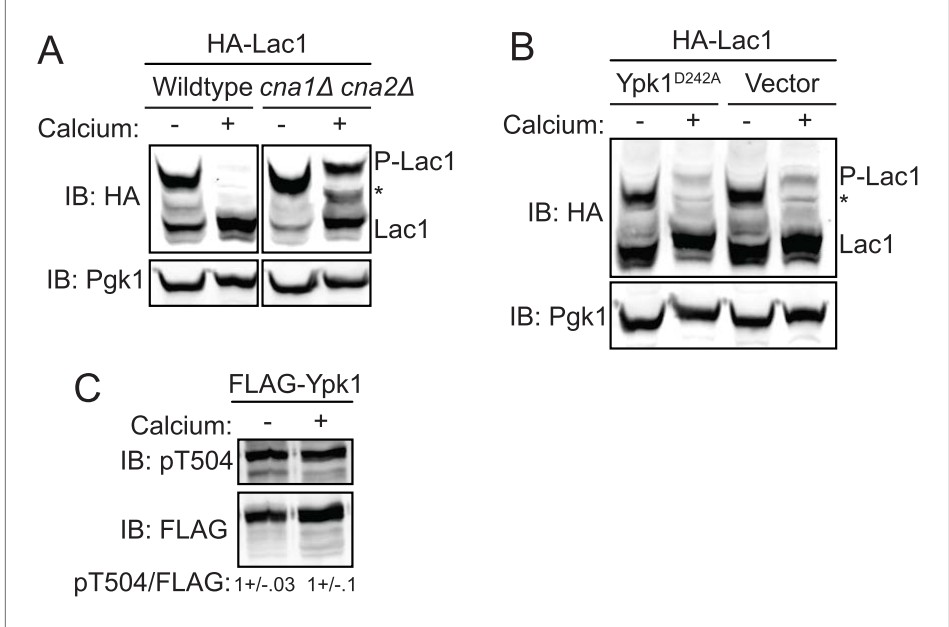

**Figure 5**. Activation of calcineurin leads to rapid dephosphorylation of Lac1 and Lag1 without affecting Ypk1 function. (**A**) Wildtype (BY4741) or *cna1Δ cna2Δ* (JTY5574) strains were transformed with a centromeric plasmid encoding 3xHA-Lac1 expressed under control of its endogenous promoter (pAX136). Cultures were grown to mid-exponential phase in selective media and then treated with 200 mM CaCl$_2$ for 10 min to activate calcineurin. Cultures were then harvested and whole cell extracts were prepared from each sample, resolved by Phos-tag SDS-PAGE, and immunoblotted as in *Figure 3*. (**B**) 3xHA-Lac1 (yAM165-A) cells were transformed with a centromeric plasmid encoding the hyperactive TORC2 independent Ypk1$^{D242A}$ allele expressed under its own promoter (pFR273) or vector (pRS316). Cultures were grown in selective media to mid-exponential phase before treatment with 200 mM CaCl$_2$ for 10 min. Cultures were then harvested and whole cell extracts were prepared from each sample, resolved by Phos-tag SDS-PAGE, and immunoblotted as above. (**C**) 3xFLAG-Ypk1 (YDB379) cells were grown to mid-exponential phase in YPD. Cultures were treated with or without 200 mM CaCl$_2$ for 10 min. Cells were then harvested and whole extracts prepared. Extracts were resolved by SDS-PAGE and blotted with anti-pSGK(T256), which recognizes Ypk1 T504 activation loop phosphorylation (*Roelants et al., 2010*) and anti-FLAG antibody (Ypk1 loading control). The blot is representative of triplicate samples and the quantitation of the ratio of pT504/FLAG from these replicates is shown below the blot.

(*Mulet et al., 2006*; *Daquinag et al., 2007*; *Berchtold et al., 2012*). Hence, it was possible, therefore, that the observed loss of phospho-Lac1 and -Lag1 might be due to a Ca$^{2+}$-stimulated and calcineurin-mediated reduction in their TORC2-Ypk1-dependent phosphorylation, rather than to direct action of calcineurin on phospho-Lac1 and -Lag1. To rule out the former possibility, we examined Lac1 and Lag1 phosphorylation in cells expressing constitutively-active Ypk1(D242A), which we have demonstrated bypasses the need for its TORC2-dependent activation (*Roelants et al., 2011*). We found that Ca$^{2+}$ addition still led to nearly complete Lac1 and Lag1 dephosphorylation in these cells (*Figure 5B*). Furthermore, as judged by immunoblotting with a phospho-site specific antibody, Pkh1- and Pkh2-dependent phosphorylation of the activation loop in Ypk1 (*Figure 5C*) was not diminished in cells stimulated with Ca$^{2+}$, confirming that there was no calcineurin-mediated decrease in the amount of active Ypk1 present. Therefore, it seems clear that calcineurin dephosphorylates the sites in Lac1 and Lag1 phosphorylated by Ypk1 directly, rather than through down-regulation of Ypk1 function.

## Ypk1-mediated phosphorylation of Lac and Lag1 stimulates ceramide synthase activity

Given the phenotypic evidence that TORC2-Ypk1-dependent modulation of Lac1 and Lag1 is physiologically important (*Figure 4C*), we next conducted biochemical analysis to determine how Ypk1-mediated phosphorylation affects the function of Lac1 and Lag1 in ceramide synthesis. Toward that

end, we first used LC-MS to monitor the levels of LCBs and LCBPs extracted from equivalent numbers of cells from exponentially-growing cultures of wild-type cells or otherwise isogenic cells expressing as the sole source of Lac1 and Lag1 either the non-phosphorylatable Lac1(S23A S24A) and Lag1(S23A S24A) mutants or the phosphomimetic Lac1(S23E S24E) and Lag1(S23E S24E) mutants. Relative to wild-type cells, the Lac1(S23A S24A) Lag1(S23A S24A) cells accumulated significantly more PHS (*Figure 6A*, *top left*), as well as more dihydrosphingosine (DHS) (*Figure 6A*, *top right*), which is a more minor LCB in yeast (note the difference in scale of the ordinate), whereas the Lac1(S23E S24E) Lag1(S23E S24E) cells displayed a level of both PHS and DHS that was somewhat lower, and a similar trend was observed even for PHS-1P, an even less abundant metabolite (*Figure 6A*, *bottom left*). The level of DHS-1P (*Figure 6A*, *bottom right*) was so low as to make reliable measurement difficult, but no differences between strains could be detected. Given that the ceramide synthase complex is responsible for the conversion of LCBs into ceramides, these findings indicate that, in the absence of Ypk1-mediated phosphorylation, the rate of LCB utilization is significantly reduced, consistent with the conclusion that Ypk1-dependent modification of Lac1 and Lag1 promotes ceramide synthase function.

As an independent means to measure flux through the sphingolipid pathway, and because all complex sphingolipids in yeast contain inositol-phosphate, an equivalent number of cells of the same three strains in mid-exponential phase were pulse-labeled, in triplicate, with $[^{32}P]PO_4^{-3}$, and the acidic sphingolipids extracted and analyzed by thin-layer chromatography. Strikingly, the amount of complex sphingolipids generated during the pulse was reproducibly higher in the Lac1(S23E S24E) Lag1(S23E S24E) cells than in the wild-type controls, and Lac1(S23A S24A) Lag1(S23A S24A) generated levels of complex sphingolipids lower than the wild-type controls (*Figure 6B*). These findings are again consistent with the conclusion that Ypk1-mediated phosphorylation of Lac1 and Lag1 stimulates the production of the ceramide precursors to complex sphingolipids.

Ypk1-mediated phosphorylation could stimulate the ceramide synthase reaction in vivo by stabilizing Lac1 and Lag1 thereby increasing their steady-state level, by enhancing their association with the small ancillary subunit Lip1, and/or by direct activation. Immunoblotting of exponentially-growing cultures expressing the 3xHA-tagged versions of wild-type Lac1 and Lag1 and the Lac1(S23A S24A) Lag1(S23A S24A) and Lac1(S23E S24E) Lag1(S23E S24E) indicated no discernible difference in their steady-state level (see *Figure 4D*). Likewise, in cells co-expressing the same proteins and FLAG-tagged Lip1 (gift of Howard Riezman, Univ. of Geneva), we observed no difference in the efficiency of Lip1 co-immunoprecipitation between wild-type Lac1 and Lag1 and either the Lac1(S23A S24A) Lag1(S23A S24A) or Lac1(S23E S24E) Lag1(S23E S24E) mutants (data not shown). These results suggested that Ypk1-mediated phosphorylation may directly enhance the catalytic efficiency of Lac1 and Lag1. To test this possibility directly, 3xFLAG-tagged versions of Lac1 and Lag1 were immunopurified from detergent-solubilized microsomes isolated from exponentially-growing cells and equivalent amounts of the resulting protein assayed in vitro, monitoring the formation of ceramide from PHS and steroyl-CoA by LC-MS. No product was observed in the absence of added steroyl-CoA (data not shown), and product formation was reduced by 85% in the presence of PHS and steroyl-CoA if 1 µM australifungin, a demonstrated and specific ceramide synthase inhibitor (*Mandala and Harris, 2000*), was added (data not shown), confirming that the reaction measured was catalyzed by ceramide synthase. We found that the specific activity of the Lac1(S23E S24E) Lag1(S23E S24E) complex was reproducibly ~2 higher than either the Lac1(S23A S24A) Lag1(S23A S24A) mutant or the wild-type complex (*Figure 6C*, *upper panel*), similar to the degree of difference in sphingolipid metabolites between these same strains that we measured by other means (*Figure 6A,B*). In two independent trials (each performed in triplicate), the identical trend was found when microsomes from these same cells were assayed directly (i.e. without detergent solubilization and enzyme enrichment by immunoprecipitation [data not shown]). Additionally, we found that immunopurified ceramide synthase from wild-type cultures treated with myriocin had higher activity than ceramide synthase prepared from untreated cells (*Figure 6C*, *lower panel*). Furthermore, this increase in ceramide synthase activity in response to myriocin treatment was not observed in Lac1(S23A S24A) Lag1(S23A S24A) cells (*Figure 6C*, *lower panel*), consistent with TORC2-Ypk1 signaling increasing ceramide synthase activity by phosphorylation at these residues. These findings are also in agreement with a reported ~two-fold decrease in the rate of ceramide production by ceramide synthase complex isolated from TORC2-deficient yeast (*Aronova et al., 2008*). Hence, we conclude that Ypk1 phosphorylation directly increases the catalytic activity of ceramide synthase.

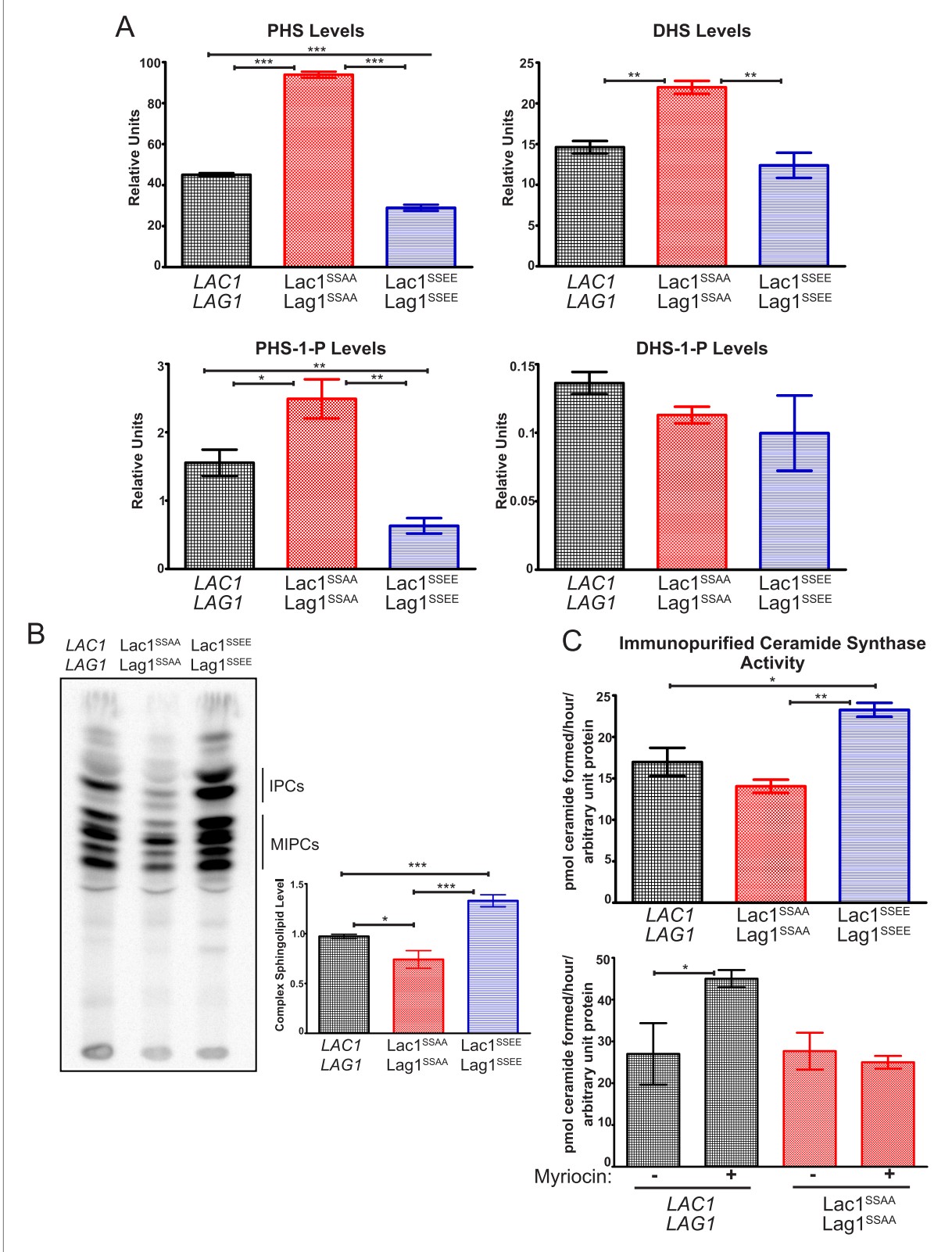

**Figure 6**. Ypk1 phosphorylation of Lac1 and Lag1 stimulates ceramide synthase activity. (**A**) Cultures of *LAC1 LAG1* (yAM205–A), Lac1^SSAA Lag1^SSAA (yAM207B) and Lac1^SSEE Lag1^SSEE (yAM210) strains were grown to mid-exponential phase and then harvested. Sphingolipids were extracted and analyzed as described in 'Materials and methods'. Values represent the mean of three independent experiments (each performed in triplicate) and error bars

*Figure 6. Continued on next page*

*Figure 6. Continued*

represent SEM. (**B**) Triplicate exponentially-growing cultures of *LAC1 LAG1* (yAM205–A), Lac1$^{SSAA}$ Lag1$^{SSAA}$ (yAM207–B) and Lac1$^{SSEE}$ Lag1$^{SSEE}$ (yAM210) were grown overnight and duplicate cultures were diluted to OD$_{600}$ = 1.0. Complex sphingolipids were labeled and analyzed by thin layer chromatography (TLC) as in 'Materials and methods'. A representative TLC plate is shown with the origin at the bottom of the image. The assigned identity of species as IPCs and MIPCs was confirmed by pharmacological or genetic inhibition of the production of these species in control cultures (data not shown). Quantification of total complex sphingolipids was performed in ImageJ by integrating the Phosphorimager screen intensity across the lane for each sample and normalized to 1 for the wild-type ceramide synthase samples. (**C**) *Upper*, ceramide synthase was purified by FLAG immunopurification from 3xHA-Lac1::*HIS3* 3xFLAG-Lag1::*LEU2* (yAM168) 3xHA-Lac1(S23A S24A)::*HIS3* 3xFLAG-Lag1(S23A S24A)::*LEU2* (Lac1$^{SSAA}$ Lag1$^{SSAA}$) (yAM184) and 3xHA-Lac1(S23E S24E)::*HIS3* 3xFLAG-Lag(S23E S24E)1::*LEU2* (Lac1$^{SSEE}$ Lag1$^{SSEE}$) (yAM192–A) cells. Immunoprecipitates were then split into three fractions and in vitro ceramide synthase assays (60 min reactions) were performed in triplicate. A small sample of each ceramide synthase assay was resolved by SDS-PAGE and immunoblotted. The signal intensity quantified from the immunoblot was then used to normalize ceramide synthase activity in each sample. *Lower*, ceramide synthase was immunopurified from 3xHA-Lac1::*HIS3* 3xFLAG-Lag1::*LEU2* (yAM168) 3xHA-Lac1(S23A S24A)::*HIS3* or 3xFLAG-Lag1(S23A S24A)::*LEU2* (Lac1$^{SSAA}$ Lag1$^{SSAA}$) (yAM184) cells as above except cultures were treated with 1.0 μM myriocin or methanol (vehicle) prior to harvesting. Values represent the mean of three independent experiments (each performed in triplicate) and error bars represent SEM. Statistical significance of values (Student's *t* test): *p = <0.05, **p = <0.009; and, ***p < 0.0009.

## Ypk1-mediated stimulation of Lac1 and Lag1 prevents autophagy induction during TORC2-driven up-regulation of sphingolipid biosynthesis

The observed increase in ceramide synthase activity in response to Ypk1-mediated phosphorylation could serve two roles that are not mutually exclusive: (i) to produce more ceramide and the derived complex sphingolipids; and, (ii) to prevent inadvertent accumulation of LCBs and the derived LCBPs when TORC2-driven Ypk1 activation stimulates metabolic flow into the sphingolipid pathway by alleviating Orm1- and Orm2-imposed inhibition of SPT (*Roelants et al., 2011*). We reasoned that if the latter were one of the important physiological functions of Ypk1-dependent stimulation of ceramide synthase, then TORC2 activation would be detrimental to cells in which the Lac1(S23A S24A) Lag1(S23A S24A) mutant is the sole source of this enzyme. To mimic TORC2-stimulated elevation of Ypk1 activity, the constitutively-active Ypk1(D242A) allele (hereafter referred to as Ypk1*) was expressed from the *YPK1* promoter on a *CEN* plasmid in either wild-type cells or the Lac1(S23A S24A) Lag1(S23A S24A) mutant. Indeed, compared to the empty vector control, expression of Ypk1* was well tolerated by cells containing wild-type Lac1 and Lag1, but deleterious to the growth of the Lac1(S23A S24A) Lag1(S23A S24A) mutant cells, whether measured on agar plates (*Figure 7A*, top) or in liquid culture (*Figure 7A*, bottom).

Given that accumulation of LCBPs has been shown to be toxic to yeast cell growth (*Kim et al., 2000*), we reasoned that the most likely metabolic perturbation responsible for the observed decrease in growth in the Lac1(S23A S24A) Lag1(S23A S24A) cells expressing Ypk1* was the build-up of LCBs and derived LCBPs. Consistent with this conclusion, we found a reproducible and statistically significant increase in the LCBP level in Lac1(S23A S24A) Lag1(S23A S24A) cells, compared to *LAC1⁺ LAG1⁺* controls, and a further increase in Lac1(S23A S24A) Lag1(S23A S24A) cells expressing Ypk1* (*Figure 7B*). If accumulation of LCBPs in Lac1(S23A S24A) Lag1(S23A S24A) co-expressing Ypk1* is indeed responsible for the poor growth, then reduction in LCBP synthesis by elimination of the gene *LCB4*, which encodes the major LCB kinase (*Nagiec et al., 1998*), should alleviate the growth inhibition. As expected, introduction of an *lcb4Δ* null mutation into the Lac1(S23A S24A) Lag1(S23A S24A) mutant suppressed the growth inhibitory effect of Ypk1* (*Figure 7C*). Conversely, and consistent with toxicity arising from accumulation of LCBPs when Lac1 and Lag1 cannot be stimulated by Ypk1, we found that the poor growth phenotype of cells lacking the gene (*LCB3*) encoding the major LCBP phosphatase (*Mandala et al., 1998*), was markedly exacerbated by introduction of the Lac1(S23A S24A) Lag1(S23A S24A) alleles, but not by introduction of the Lac1(S23E S24E) Lag1(S23E S24E) alleles (*Figure 7D*). In fact, and strikingly, presence of the Lac1(S23E S24E) Lag1(S23E S24E) allele afforded nearly complete rescue of the slow-growth phenotype of the *lcb3Δ* mutation (*Figure 7D*). Consistent with the toxicity of LCBPs when LCB utilization by ceramide synthase is inefficient, an *elo3* mutation, which prevents efficient formation of C$_{26}$-CoA (an acyl chain found in yeast ceramides), was synthetically lethal with *lcb3Δ* (*Kobayashi and Nagiec, 2003*).

Finally, it has been reported that aberrant increases in LCBP level impede growth by triggering autophagy even under nutrient-rich conditions (*Zimmermann et al., 2013*). In agreement with the

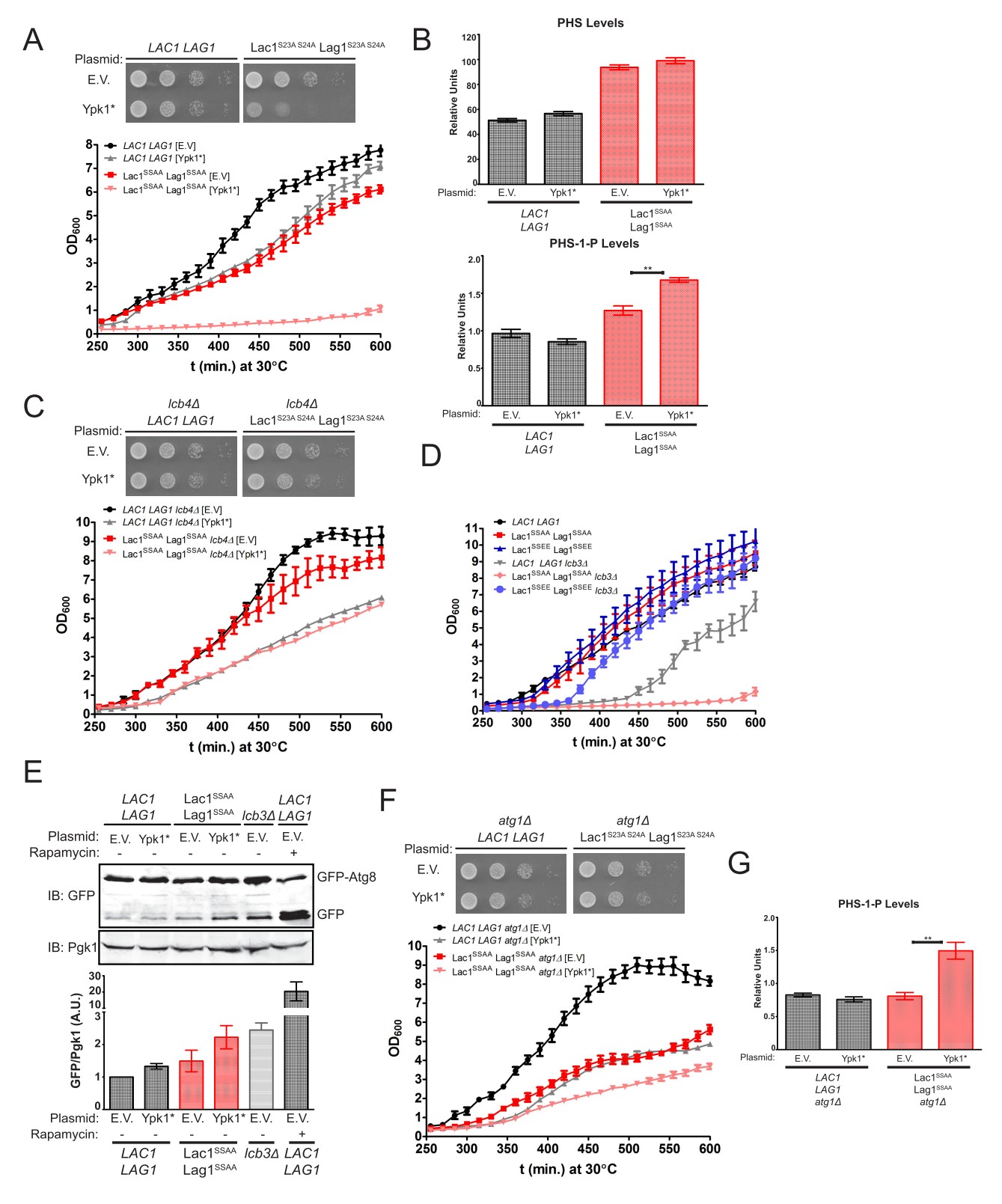

**Figure 7**. Failure of Ypk1 to upregulate ceramide synthase causes LCBP accumulation that triggers autophagy. (**A**) *LAC1 LAG1* (yAM205–A) or Lac1$^{SSAA}$ Lag1$^{SSAA}$ (yAM207–B) were transformed with P$_{YPK1}$-Ypk1$^{D242A}$ (shown as Ypk1*) (pFR273) or empty vector pRS316 (EV). Transformants were grown to exponential phase in synthetic complete medium and then diluted to OD$_{600}$ = 0.1 and grown in microtiter plates (*lower*) or on agar plates (*upper*). For *Figure 7. Continued on next page*

*Figure 7. Continued*

liquid cultures, each was grown in at least quadruplicate and the error bars indicate the SEM of replicates at each time point. (**B**) Cells from (**A**) were grown to mid-exponential phase in selective synthetic complete media and then harvested. Sphingolipids were extracted and analyzed as described in 'Materials and methods'. Values represent the mean of three independent experiments (each performed in triplicate) and error bars represent SEM. (**C**) *LAC1 LAG1 lcb4Δ* (yAM237) or Lac1^SSAA Lag1^SSAA *lcb4Δ* (yAM238–A) were transformed with Ypk1* (pFR273) and growth experiments performed as in (**A**). (**D**) Liquid growth assays were performed as in (**A**) for *LAC1 LAG1* (yAM205–A), Lac1^SSAA Lag1^SSAA (yAM207–B), Lac1^SSEE Lag1^SSEE (yAM210), *LAC1 LAG1 lcb3Δ* (yGT12), Lac1^SSAA Lag1^SSAA *lcb3Δ* (yGT13) and Lac1^SSEE Lag1^SSEE *lcb3Δ* (yGT14) strains. (**E**) *LAC1 LAG1* (yAM205–A) or Lac1^SSAA Lag1^SSAA (yAM207–B) or *LAC1 LAG1 lcb3Δ* (yGT12) strains were transformed with Ypk1* (pFR273) or pRS316 (EV) and additionally P$_{TPI1}$-GFP-Atg8. Growing cultures treated with vehicle or 2 μg/ml rapamycin for 2 hr and then harvested and whole extracts prepared. Extracts were resolved by SDS-PAGE and blotted with anti-GFP to detect GFP-Atg8 and free GFP arising from GFP-Atg8 autophagic processing and anti-Pgk1 antibody. The blot is representative of triplicate samples and the quantitation of the ratio of free GFP/Pgk1 from these replicates is shown below the blot. (**F**) *LAC1 LAG1 atg1Δ* (yAM239–A) or Lac1^SSAA Lag1^SSAA *atg1Δ* (yAM240–A) sensitivity to Ypk1* was measured as in (**A**). (**G**) Cells from (**F**) were grown to mid-exponential phase in selective synthetic complete media and then harvested. Sphingolipids were extracted and analyzed as described in (**B**).

poor growth arising from LCBP-evoked induction of autophagy, we found, first, that Lac1(S23A S24A) Lag1(S23A S24A) cells expressing Ypk1* exhibited a readily detectable increase in basal GFP-Atg8 processing, comparable to that in *lcb3Δ* cells (but, of course, much less than that occurring when cells were treated with the starvation mimetic rapamycin) (*Figure 7E*). Second, we found that preventing autophagy by ablating the gene *ATG1*, which encodes a protein kinase necessary for induction of autophagophore formation and its elongation (*Papinski and Kraft, 2014*), provided substantial rescue of the growth debilitating effect of Ypk1* expression in Lac1(S23A S24A) Lag1(S23A S24A) cells (*Figure 7F*, compare to *Figure 7A*). This rescue was not due to an indirect effect of the absence of Atg1 on LCBP level because introduction of the *atg1Δ* mutation did not prevent the observed hyper-accumulation of LCBP in Lac1(S23A S24A) Lag1(S23A S24A) cells expressing Ypk1* (*Figure 7G*). Collectively, these results indicate that, aside from stimulating ceramide synthesis per se, another physiologically important role of Ypk1-dependent ceramide synthase activation is, at least in part, to prevent hyper-accumulation of LCBPs and thereby avoid inadvertent induction of autophagy under nutrient-sufficient conditions (*Figure 8*).

## Discussion

### Identification of new TORC2-Ypk1 substrates

Our approach identified Lac1 and Lag1 as potential Ypk1 targets, and our subsequent characterization demonstrated unequivocally that both Lac1 and Lag1 are *bona fide* Ypk1 substrates in vivo and that their Ypk1-dependent modification is biologically important for optimal modulation of sphingolipid metabolism. These findings validate the ability of our methods for discovery of physiologically relevant protein kinase substrates. Phospho-acceptor site motif pattern-matching alone, although useful in identifying kinase substrates in some cases (*Yaffe et al., 2001*; *Manning et al., 2002*; *Mah et al., 2005*; *Holt et al., 2007*; *Linding et al., 2007*; *Rennefahrt et al., 2007*; *Gwinn et al., 2008*; *Hutti et al., 2009*), can yield a large number of false positives. This possibility was a significant concern for us because certain features of the basophilic Ypk1 motif are shared with other protein kinases (*Mok et al., 2010*). As a means to avoid this problem, we devised a novel SDL-based genetic approach to apply as a secondary filter to parse the bioinformatically selected candidates further.

Although not the only possible explanation for detecting an SDL hit, one mechanism our genetic method should be able to assess is whether the candidate gene product displays a primary characteristic of a true substrate, namely the ability to compete with other substrates for association with Ypk1. A genuine Ypk1 substrate should, when highly over-expressed, sequester a large fraction of the available pool of active kinase, and thus impede its actions on its essential cellular targets. In the absence of Ypk2, over-expression of an authentic Ypk1 substrate should therefore be deleterious for growth when the activity of an analog-sensitive Ypk1 allele is reduced with a selective inhibitor. The fact that 70% of the SDL hits were indeed substrates for Ypk1-mediated phosphorylation in vitro verified that our use of this genetic method as a secondary filter to pick out true substrates from the list of bioinformatically identified candidates was well justified. In this regard, overexpression of Fpk1(KD), a catalytically-inactive non-essential Ypk1 substrate, yielded a readily detectable SDL phenotype. Thus,

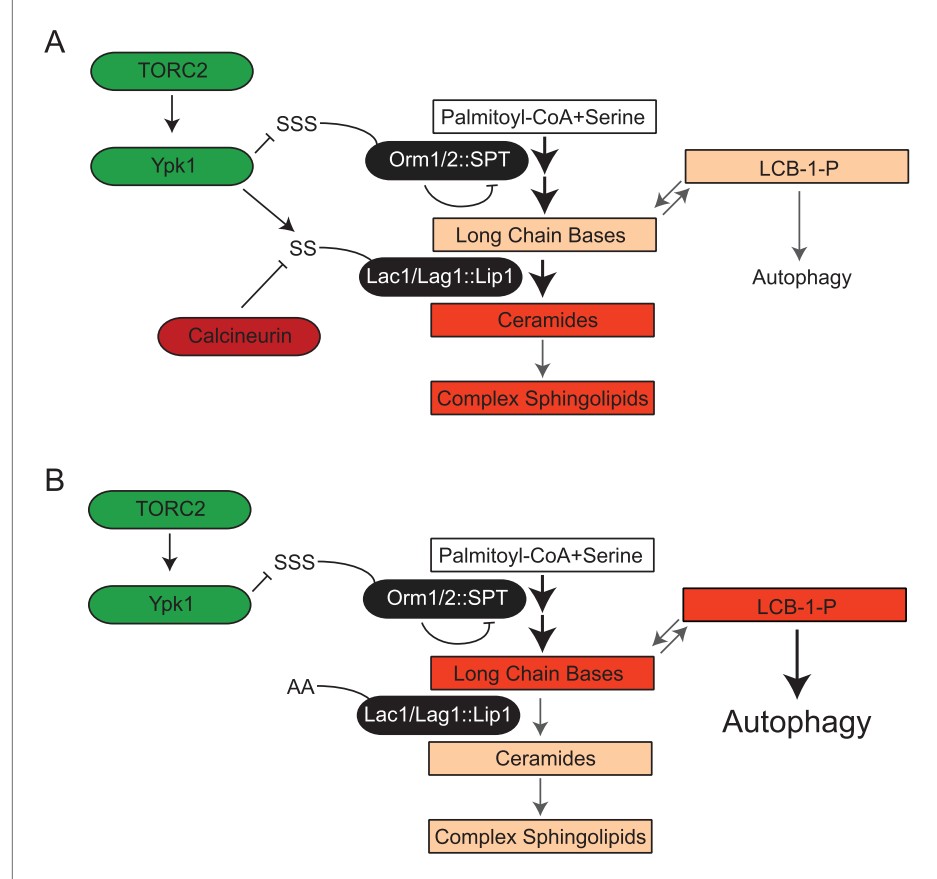

**Figure 8**. TORC2-Ypk1 signaling globally activates sphingolipid synthesis, selectively directs flux toward ceramide metabolites, and prevents LCBP cross-talk to the autophagy pathway. (**A**) Diagram of yeast de novo sphingolipid biosynthesis shown is derived from (***Dickson, 2008***). Enzymes are in ovals. Metabolites are in boxes. Increased color intensity indicates level of metabolite increase in response TORC2-Ypk1 activation. TORC2-Ypk1 signaling globally activates de novo sphingolipid biosynthesis via derepression of the SPT complex (***Roelants et al., 2011***; ***Berchtold et al., 2012***; ***Sun et al., 2012***), potentially increasing levels of all metabolites. However, Ypk1 also upregulates ceramide synthesis via phosphorylation of Lac1 and Lag1, thus primarily directing this increased flux towards ceramides and away from LCBs and LCBPs. (**B**) In the absence of Ypk1 mediated ceramide biosynthesis regulation, increased sphingolipid flux raises LCB and LCBP levels. This slows cell growth by activating autophagy. Thus, TORC2-Ypk1 signaling not only activates sphingolipid biosynthesis in response to stress, but also insulates this flux towards ceramides to prevent metabolite mediated crosstalk to the autophagy machinery.

using the protocol we devised, SDL may be a generally useful way to identify substrates and perhaps other binding partners of protein kinases, not just those directly connected to any output phenotype being measured.

Other genetic approaches have also been useful in identifying physiologically relevant targets of Ypk1. For example, a transposon insertion that suppressed the growth defect of a *ypk1-ts ypk2Δ* strain at an otherwise non-permissive temperature initially identifed Orm2 as a potential Ypk1 substrate (***Roelants et al., 2002***), a finding later corroborated by us (***Roelants et al., 2011***) and others (***Berchtold et al., 2012***; ***Liu et al., 2012***; ***Niles et al., 2012***; ***Sun et al., 2012***). Similarly, Smp1, which we confirmed here is a likely Ypk1 substrate, was first identified as a potential Ypk1 target because it was isolated as a dosage suppressor of the temperature-sensitive phenotype of *ypk1-ts ypk2Δ* cells (***Roelants et al., 2002***). Smp1 is a transcription factor (***de Nadal et al., 2003***) that mediates iron toxicity (***Lee et al., 2012a***) and, similarly, Ypk1 is required for iron toxicity (***Lee et al., 2012a***). Thus, TORC2-Ypk1 signaling might be mechanistically coupled to iron metabolism by modulation of Smp1 transcriptional output.

Likewise, other situations where chemical genetics can be applied have proven useful in gleaning what aspects of cell function are controlled by a protein kinase. For example, mutagenesis of yeast Tor2 to confer susceptibility to a chemical inhibitor and thereby selectively inhibit TORC2 action has been achieved recently (*Kliegman et al., 2013*). This tool was combined with a collection of deletion mutants to identify what processes, when eliminated, are especially deleterious to cell growth and survival when TORC2 action (and presumably Ypk1 activity) is limiting. This analysis suggested some connection between TORC2 action and the pentose-phosphate pathway (*Kliegman et al., 2013*), in keeping with the growth-promoting roles of both TORC1 and TORC2 and the demand for NADPH in many cellular anabolic reactions. Similarly, use of TOR inhibitors has implicated TORC2-Ypk1 signaling in regulation of actin filament formation that is somehow required for yeast cell survival in response to low levels of DNA damage (*Shimada et al., 2013*).

By contrast, the chemical genetic approach in our SDL method is quite different, in that it scores the deleterious effect arising from overexpression of a gene product (increased protein dosage) rather than from the total absence of a gene product, when activity of the kinase of interest is limited by inhibitor. Theoretically, under our conditions, we should also have been able to observe synthetic dosage rescue ('SDR'); however, we found no such examples. In any event, our method independently identified nearly all of the previously known in vivo substrates of Ypk1, as well as nearly a dozen genuine Ypk1 targets, including Lac1 and Lag1, that have only been pinpointed by our three-tiered method. Thus, our SDL screening technique provides a complementary approach for identifying substrates of a protein kinase above and beyond those accessible through genetic interactions between the kinase and single-gene deletions or other genetic schemes.

Many of the gene products identified by our screen are involved in processes that the TORC2-Ypk1 signaling axis is already known to regulate. For example, Ypk1 regulates glycerol-3-phosphate production via phosphorylation and inhibition of glycerol-3-phosphate dehydrogenase Gpd1 (*Lee et al., 2012b*). Interesting, a very likely Ypk1 target that met all of the criteria in our screen is Gpt2, sn-glycerol-3-phosphate 1-acyltransferase (*Zheng and Zou, 2001*), an enzyme that esterifies glycerol-3-phosphate as the first step in glycerolipid formation. In this same regard, as another very likely Ypk1 target we also identified Fps1, a membrane channel (aquaglyceroporin) that regulates efflux of glycerol (a glycerol-3-phosphate-derived metabolite) (*Luyten et al., 1995*). These results strengthen the conclusion that TORC2-Ypk1 signaling is intimately involved in modulating the level of the precursor to both glycerophospholipids and the osmolyte glycerol (*Lee et al., 2012b*).

Ypk1 action has been implicated in regulation of both fluid phase and receptor-mediated endocytosis (*deHart et al., 2002*). In this regard, we identified as a very likely Ypk1 target an endocytic adaptor, the α-arrestin Rod1, which is necessary for ubiquitinylation-triggered internalization of nutrient permeases (*Lin et al., 2008*; *Becuwe et al., 2012*) and the pheromone receptor Ste2 (*Alvaro et al., 2014*). Thus, as it does for PM lipids, TORC2-Ypk1 signaling may modulate PM protein composition via this α-arrestin, a possibility we are pursuing.

Ypk1 function has also been implicated in regulating production of reactive oxygen species (ROS) (*Niles et al., 2014*), but an as yet undefined mechanism. In our screen, we found Ysp2, a protein that regulates mitochondrial morphology and ROS levels (*Sokolov et al., 2006*), as a very likely Ypk1 substrate, possibly providing insight into the molecular basis of the connection between TORC2-Ypk1 signaling and ROS levels.

Similarly, several other prospects that were identified by our screen as very likely Ypk1 substrates remain to be validated. Such candidates include Muk1, a GEF for yeast Rab 5-type small GTPases (Vps21, Ypt52, and Ypt53) involved in vesicle-mediated Golgi body-to-endosome trafficking (*Paulsel et al., 2013*), suggesting that Ypk1 may also control switches that direct the flow of lipids. Muk1 is also intriguing for another potential reason. In *Schizosaccharomyces pombe*, a Rab 5-like GTPase (Ryh1) and its Muk1-like GEF were identified in a screen for TORC2 activators (*Tatebe et al., 2010*). Thus, if Ypk1-mediated phosphorylation inhibits Muk1 function, it could represent a negative feedback mechanism exerted on TORC2; conversely, if Ypk1-mediated phosphorylation stimulates Muk1 function, it could represent a mechanism for self-reinforcing maintenance of TORC2 activity and, thus, a high level of activated Ypk1. Clearly, by further investigating the physiological relevance of these and other remaining candidates much new biology may be learned. Indeed, several gene products of totally unknown function, such as Yhr097c and Ynr014w, as well as gene products (e.g., Atg21 and Pex31) not previous linked to either TORC2 or Ypk1, if validated, may provide new mechanistic insight into additional cellular processes regulated by TORC2-Ypk1 signaling.

## Mechanism of phosphoregulation of ceramide synthase

As we have demonstrated here, TORC2-dependent Ypk1-mediated phosphorylation of Lac1 and Lag1 stimulates the function of the ceramide synthase complex. Consistent with our findings, a previous study found a consistent decrease in ceramide synthase activity in microsomal fractions isolated from cells in which TORC2 had been inactivated (and thus Ypk1 activity was presumably reduced) (*Aronova et al., 2008*), although indirect effects of the loss of TORC2 function on ceramide synthase activity could not be ruled out. Our findings make it clear that the role of TORC2 is to promote the Ypk1-dependent phosphorylation of Lac1 and Lag1 subunits of this enzyme. However, the precise molecular mechanism by which this post-translational modification stimulates this enzyme is still not completely clear. In this regard, it has been shown that mammalian ceramide synthase activity increases upon heterodimerization of different catalytic subunit isoforms (*Laviad et al., 2012*). However, as judged by co-immunoprecipitation, we found no difference in the state of Lac1-Lag1 association between the wild-type proteins and either our Lac1(S23A S24A) Lag1(S23A S24A) or Lac1(S23E S24E) Lag1(S23E S24E) mutants (data not shown). As mentioned in Results, we found no difference in the steady-state level of these same complexes or in their content of Lip1, a non-catalytic component of the complexes also essential for ceramide synthase activity (*Vallée and Riezman, 2005*). Thus, understanding of how phosphorylation of Ser23 and Ser24 in Lac1 and Lag1 stimulate ceramide synthase activity may require detailed structural information, which will be challenging to obtain for these polytopic integral membrane proteins.

## TORC2-Ypk1 control of ceramide synthesis is conserved

Although the enzymic steps that carry out sphingolipid biosynthesis have been largely elucidated, much less was known, until recently, about regulation of these enzymes (*Breslow and Weissman, 2010*; *Breslow, 2013*). The first insight came when it was demonstrated (*Roelants et al., 2011*; *Berchtold et al., 2012*; *Sun et al., 2012*) that, in response to PM stresses, including treatment with myriocin and aureobasidin A, TORC2-Ypk1 signaling is activated and alleviates inhibition of the SPT complex by phosphorylating the negative regulatory proteins Orm1 and Orm2 (*Figure 8*). As a direct consequence, the rate of de novo production of the LCB precursor to sphingolipids is increased. Although TORC2 signaling had been implicated in promoting synthesis of ceramide, the product of LCB N-acylation (*Aronova et al., 2008*), it was unknown whether that role was simply the result of TORC2-Ypk1-dependent stimulation of SPT function and the resulting increase in LCB supply. As we demonstrated here, Ypk1-mediated phosphorylation of Lac1 and Lag1 also increases in response to both myriocin and aureobasidin A, suggesting that ceramide synthesis per se, and not simply general elevation of LCB levels, is important for allowing the cells to cope with the effects of these antibiotics. Indeed, collectively, the findings we describe here demonstrate unequivocally that, in addition to up-regulation of SPT, TORC2-Ypk1 exerts direct control on the ceramide synthase step of the sphingolipid biosynthetic pathway by phosphorylating and stimulating the function of the Lac1 and Lag1 subunits of the ceramide synthase complex. Moreover, as we also demonstrated, the ceramide synthase reaction represents an important branch point in sphingolipid biosynthesis (*Figures 2B and 8*). LCBs produced by the SPT reaction can either be converted to ceramides or become phosphorylated by LCB kinase Lcb4 (and its paralog Lcb5) to form LCBPs (*Nagiec et al., 1998*). Accumulation of LCBPs has been shown to be toxic to yeast cell growth (*Kim et al., 2000*), as we have also confirmed here, at least in large part because, as is now known, these metabolites trigger inappropriate induction of autophagy (*Zimmermann et al., 2013*). Thus, the rate of ceramide production must be properly adjusted to maintain the pool of LCBs and derived LCBPs at a non-deleterious level, in agreement with evidence in yeast and other organisms that ceramides and LCBPs generally play antagonist roles and must be maintained in the proper dynamic balance (*Kobayashi and Nagiec, 2003*; *Spiegel and Milstien, 2003*; *Kihara et al., 2007*; *Dickson, 2008*; *Breslow and Weissman, 2010*; *Bikman and Summers, 2011*). Hence, the function we have discovered and described here for TORC2-Ypk1 in stimulating ceramide synthase promotes utilization of the increased LCB generated upon TORC2-Ypk1-mediated up-regulation of SPT. This metabolic control has multiple clear-cut physiological benefits to the cell: (a) directing flow in the sphingolipid pathway toward complex sphingolipids to populate the PM barrier; (b) reduction of the level of potentially toxic LCBPs; and, (c) avoidance of inappropriate induction of autophagy under nutrient-sufficient conditions (*Figure 8*). Indeed, our results indicate that a significant role for the coordination exerted by TORC2-Ypk1 between the level of SPT activity and the level of ceramide synthase activity is to prevent metabolic 'cross-talk' to the autophagy pathway.

If this function of TORC2-Ypk1 signaling is important, then it should be conserved. Indeed, alignments of the primary structures of Lac1 and Lag1 homologs predicted from sequenced fungal genomes, from *Saccharomyces sensu stricto* species to the very distantly related *Ustilago maydis*, nearly all contain a basophilic sequence that matches the Ypk1 phospho-acceptor site consensus and is located, as in *S. cerevisiae* Lac1 and Lag1, in their N-terminal cytosolic extensions, suggests that Ypk1-related protein kinases may regulate ceramide synthase across essentially all fungal species. Consistent with this suggestion, a genetic interaction between a Ypk1 homolog (YpkA) and a Lac1/Lag1-like ceramide synthase component (BarA1) has been reported in *Aspergillus nidulans* (***Colabardini et al., 2013***).

There is also evidence that Orm1 and Orm2, when phosphorylated at unique sites by protein kinase Npr1, promotes steps in the sphingolipid pathway that lead to more complex sphingolipids (***Shimobayashi et al., 2013***). In contrast to Ypk1, which is activated by TORC2, Npr1 is inhibited by TORC1 (***MacGurn et al., 2011***). Thus, this control mechanism will only be exerted under conditions that inactivate TORC1, such as amino acid starvation (***Loewith and Hall, 2011***), a condition that presumably requires adjustment of both PM lipid and protein composition to maximize the cell's ability to scavenge and assimilate nutrients. Under the same condition, autophagy is induced because, like Npr1, TORC1 also negatively regulates the autophagy-inducing protein kinase Atg1-Atg13 complex (***Alers et al., 2014***). Conversely, under nutrient sufficient conditions, TORC1 is active, and phosphorylates and stimulates protein kinase Sch9. Interestingly, Sch9 action should act in concert with TORC2-Ypk1 signaling to help keep the levels of LCBs and LCBPs low and ceramides high. This is likely because Sch9 promotes transcriptional repression of genes (*YDC1* and *YPC1*) that encode ceramidases and inhibits a phosphosphingolipid phospholipase C (Isc1) that hydrolyzes complex sphingolipids (***Swinnen et al., 2014***). Such complex multi-component controls may be a general feature of signaling modalities that interface with biosynthetic pathways that have intermediates, like LCBPs, that are not inocuous, but are themselves bioactive metabolites.

## Calcineurin negatively regulates ceramide synthase

Prior studies had suggested that calcineurin negatively regulates sphingolipid production via effects on the function of the ancillary TORC2 subunits, Slm1 and Slm2 (***Bultynck et al., 2006***; ***Mulet et al., 2006***; ***Tabuchi et al., 2006***; ***Daquinag et al., 2007***), although the molecular connection between Slm1 and Slm2 and sphingolipid biosynthesis was unclear. Subsequently, it was observed that, in cells lacking the regulatory subunit (Cnb1), there was an increase in $C_{26}$-containing ceramides, suggesting that calcineurin somehow antagonizes TORC2-dependent signaling (***Aronova et al., 2008***). We found that calcineurin negatively regulates ceramide synthesis by directing the dephosphorylation of Lac1 and Lag1. This $Ca^{2+}$-activated calcineurin-dependent dephosphorylation occurred even in cells expressing a TORC2-independent constitutively-active Ypk1 allele. Moreover, we showed here that calcineurin does not affect Pkh1- (and Pkh2-) mediated phosphorylation of the activation loop of Ypk1, and we demonstrated previously that presence or absence of calcineurin does not alter TORC2-mediated phosphorylation of Ypk1 or cause any substantial change in Ypk1 specific activity (***Roelants et al., 2011***). Thus, direct down-modulation of either TORC2 or Ypk1 by calcineurin cannot account for the negative regulation it exerts on sphingolipid biosynthesis. What our findings now make clear is that calcineurin negatively regulates the sphingolipid pathway at the level of ceramide synthesis, at least in large part, by direct dephosphorylation of the stimulatory phosphorylations in the ceramide synthase subunits Lac1 and Lag1 that are installed by TORC2-Ypk1 signaling. Calcineurin recognizes substrates via a docking motif (PxIxIT or variants thereof), typically also accompanied quite a distance upstream by a secondary docking site (LxVP) (***Roy and Cyert, 2009***). However, these sites can be quite degenerate; for example, the more hydrophobic variant PVIVIT is much more potent in recruiting calcineurin when it is used to replace the native 'PxIxIT' sequences in either the transcription factor Crz1 (PIISIQ) (***Roy et al., 2007***) or the endocytic adaptor Aly1 (PILKIN) (***O'Donnell et al., 2013***). In both Lac1 and Lag1, there is a similar hydrophobic sequence located at the identical position in both proteins ([355]PIVFVL[360]). Whether this or any other degenerate match represents a calcineurin-binding site remains to be determined.

In conclusion, our screening approach has provided a number of new insights into how the TORC2-Ypk1 axis modulates sphingolipid homeostasis and has uncovered a significant number of candidate substrates that will very likely shed further light on other aspects of cellular function that are regulated by TORC2-Ypk1 signaling.

**Table 2.** *Saccharomyces cerevisiae* strains used in this study

| Strain | Genotype | Source/reference |
|---|---|---|
| BY4741 | *MAT**a** his3Δ1 leu2Δ0 met15Δ0 ura3Δ0* | Research Genetics, Inc. |
| BY4742 | *MATα his3Δ1 leu2Δ0 lys2Δ0 ura3Δ0* | Research Genetics, Inc. |
| yAM135-A | BY4741 Ypk1(L424A)::*URA3-ypk2Δ*::KanMX4 | This study |
| JTY6142 | BY4741 *ypk1Δ*::KanMX4 | Research Genetics, Inc. |
| yAM120-A | BY4741 *ypk2Δ*::KanMX4 | This study |
| yAM159-A | BY4741 3xFLAG-Lag1::*LEU2* | This study |
| yAM163-A | BY4741 3xFLAG-Lag1(S23A S24A)::*LEU2* | This study |
| yAM165-A | BY4742 3xHA-Lac1::*HIS3* | This study |
| yAM166-A | BY4742 3xHA-Lac1 (S23A S24A)::*HIS3* | This study |
| JTY5574 | BY4741 *cna1Δ*::KanMX4 *cna2Δ*::KanMX4 | M.S. Cyert, Stanford Univ. |
| YDB379 | BY4741 Ypk1-3xFLAG::natNT2 | J.S. Weissman, Univ. of California, San Francisco |
| yAM205-A | BY4742 Lac1::*LEU2* Lag1::*LEU2* | This study |
| yAM207-B | BY4742 Lac1(S23A S24A)::*LEU2* Lag1(S23A S24A)::*LEU2* | This study |
| yAM210 | BY4742 Lac1(S23E S24E)::*LEU2* Lag1(S23E S24E)::*LEU2* | This study |
| yGT12 | BY4742 *LYS2*+ Lac1::*LEU2* Lag1::*LEU2* *lcb3Δ*::natNT2 | This study |
| yGT13 | BY4742 *LYS2*+ Lac1(S23A S24A)::*LEU2* Lag1(S23A S24A)::*LEU2* *lcb3Δ*::natNT2 | This study |
| yGT14 | BY4742 *LYS2*+ Lac1(S23E S24E)::*LEU2* Lag1(S23E S24E)::*LEU2* *lcb3Δ*::natNT2 | This study |
| yAM168 | BY4741 3xHA-Lac1::*HIS3* 3xFLAG-Lag1::*LEU2* | This study |
| yAM184 | BY4741 3xHA-Lac1(S23A S24A)::*HIS3* 3xFLAG-Lag1 (S23A S24A)::*LEU2* | This study |
| yAM192-A | BY4741 *MET15*+ 3xHA-Lac1(S23E S24E)::*HIS3* 3xFLAG-Lag1(S23E S24E)::*LEU2* | This study |
| yKL4 | BY4741 *TOR2*+::Hyg^r | Kristin Leskoske, this lab |
| yKL5 | BY4741 Tor2(L2178A)::Hyg^r | Kristin Leskoske, this lab |

## Materials and methods

### Construction of yeast strains

All *S. cerevisiae* strains used in this study are listed in *Table 2*. Strains were constructed using standard yeast genetic manipulations (*Burke et al., 2005*). For all strains constructed, integration of the desired DNA fragment into the correct genomic loci was confirmed by PCR using an oligonucleotide complementary to the integrated DNA fragment and an oligonucleotide complementary to genomic sequence at least 150 bases away from the integration site.

### Plasmids and recombinant DNA methods

All plasmids used in this study (except the library of P$_{GAL1}$ based overexpression plasmids for synthetic dosage lethality [SDL] screening) are listed in *Table 3*. All plasmids were constructed and maintained in *E. coli* using standard laboratory methods (*Green and Sambrook, 2012*). For SDL screening, the entire open reading frame of each predicted and known Ypk1 substrate was amplified by PCR from BY4741 genomic DNA and ligated into the multiple cloning site of YCpLG (*CEN*, P$_{GAL1}$, *LEU2*), generating a vector allowing galactose inducible overexpression of each substrate. All constructs generated in this study were confirmed by sequence analysis covering all promoter and coding regions in the construct.

### Bioinformatic prediction of Ypk1 substrates

A Nx20 position weight matrix defining Ypk1 phosphoacceptor site specificity was made merging previously published data sets defining Ypk1 primary sequence specificity (*Casamayor et al., 1999*;

**Table 3.** Plasmids used in this study

| Plasmid | Description | Source/reference |
|---|---|---|
| pGEX6P-1 | GST tag, bacterial expression vector | GE Healthcare, Inc. |
| pGEX4T-1 | GST tag, bacterial expression vector | GE Healthcare, Inc. |
| YCpLG | *CEN*, *LEU2*, P$_{GAL1}$ vector | (*Bardwell et al., 1998*) |
| BG1805 | *2 µm*, *URA3*, P$_{GAL1}$, C-terminal tandem affinity (TAP) tag vector | Open Biosystems, Inc. |
| pRS313 | *CEN*, *HIS3*, vector | (*Sikorski and Hieter, 1989*) |
| pRS316 | *CEN*, *URA3*, vector | (*Sikorski and Hieter, 1989*) |
| pRS416 | *CEN*, *URA3*, vector | (*Sikorski and Hieter, 1989*) |
| pBC111 | *CEN*, *LEU2*, vector | (*Iida et al., 2007*) |
| CHp282 | pRS416 P$_{MET25}$-GFP | Chau Huynh, this laboratory |
| pLB215 | pRS416 P$_{MET25}$-Ypk1 | (*Niles et al., 2012*) |
| pAX53 | pRS416 P$_{MET25}$-Ypk1(K376A) | This study |
| pAX50 | BG1805 Ypk1(L424A) | This study |
| pFR203 | pGEX4T-1 Orm1(1-85) | (*Roelants et al., 2011*) |
| pBT6 | pGEX6P-1 Fps1(1-255) | This study |
| pBT7 | pGEX6P-1 Fps1(531-669) | This study |
| pBT12 | pGEX6P-1 Smp1 | This study |
| pAX55 | pGEX6P-1 Lcb3(1-79) | This study |
| pAX56 | pGEX6P-1 Cdc1(1-41) | This study |
| pAX58 | pGEX6P-1 Her1(1-224) | This study |
| pAX59 | pGEX6P-1 Rts3 | This study |
| pAX62 | pGEX6P-1 Fkh1 | This study |
| pAX63 | pGEX6P-1 Yhp1 | This study |
| pAX66 | pGEX6P-1 YNR014W | This study |
| pAX67 | pGEX6P-1 YHR097C | This study |
| pAX94 | pGEX6P-1 Mds3(545-1016) | This study |
| pAX131 | pGEX4T-1 Lac1(1-76) | This study |
| pAX132 | pGEX4T-1 Lac1(1-76)(S23A S24A) | This study |
| pFR291 | pGEX4T-1 Lag1(1-80) | This study |
| pAX133 | pGEX4T-1 Lag1(1-80)(S23A S24A) | This study |
| pAX134 | pGEX6P-1 Muk1(1-305) | This study |
| pAX215 | pGEX6P-1 Cyk3 | This study |
| pAX223 | pGEX6P-1 Gpt2(1-35) | This study |
| pAX224 | pGEX6P-1 Gpt2(570-743) | This study |
| pAX225 | pGEX6P-1 Bre5 | This study |
| pAX226 | pGEX6P-1 Npr1(1-437) | This study |
| pAX227 | pGEX6P-1 Pal1 | This study |
| pAX228 | pGEX6P-1 Ysp2(97-665) | This study |
| pAX229 | pGEX6P-1 Ysp2(1072-1282) | This study |
| pAX230 | pGEX6P-1 Atg21 | This study |
| pAX231 | pGEX6P-1 Pex31(250-462) | This study |
| pAX136 | pRS313 P$_{LAC1}$-3xHA-Lac1 | This study |
| pFR273 | pRS316 P$_{YPK1}$-Ypk1(D242A) | (*Roelants et al., 2011*) |
| pAX250 | pRS313 P$_{TPI1}$-GFP-Atg8 | This study |
| pBCT-CCH1H | pBC111 P$_{TDH3}$-Cch1 | (*Iida et al., 2007*) |

*Mok et al., 2010*). This position weight matrix was then used with MOTIPS (*Lam et al., 2010*) to identify proteins with likely phosphorylated occurrences of this motif. Yeastmine (*Balakrishnan et al., 2012*) was used to identify all *S. cerevisiae* genes that showed genetic interactions with Ypk1, Ypk2, any component of the sphingolipid biosynthetic pathway or any component of known Ypk1 regulators (TORC2 and PP2A). Genes that showed significant growth phenotypes on myriocin, aureobasidin A and caspifungin (all compounds that cause severe growth phenotypes in *ypk1Δ* strains) were identified from the literature (*Hillenmeyer et al., 2008*). Lastly, MOTIPS predicted Ypk1 phosphorylation sites were compared to Phosphogrid (*Sadowski et al., 2013*) to identify those sites known to be phosphorylated in vivo. To be considered a potential Ypk1 substrate in this study, a protein had to have a myriocin, aureobasidin or caspofungin phenotype or a Yeastmine identified genetic interaction and: (a) 4 or more MOTIPS predicted sites, (b) 3 sites with at least one above a MOTIPS likelihood score of 0.7 or a Phosphogrid identified site or (c) 1–2 sites with a MOTIPS likelihood score of 0.7 and a Phosphogrid identified site. We also considered for further analysis a limited number of proteins that did not meet these criteria, but which still contained Ypk1 motifs. These are indicated in *Table 3*.

## Yeast growth assays and synthetic dosage lethality screening

For SDL screening, yAM135-A and BY4741 strains were both transformed with each SDL plasmid. Transformants were then cultured overnight in SC media containing 2% raffinose and 0.2% sucrose. 10-fold serial dilutions of overnight cultures starting from $OD_{600} = 1.0$ were then made in sterile water and spotted onto SC solid media with 2% galactose (to induce protein expression) or 2% dextrose (no protein expression). These plates also contained 1:1000 DMSO, 1 µM 3-MB-PP1 or 2 µM 3-MB-PP1 to inhibit to varying degrees Ypk1-as kinase activity in yAM135-A. Serially spotted cultures were allowed to grow in the dark at 30°C for 3 days. Plates were then scanned on a flatbed scanner and growth phenotypes were assessed and scored.

For all SDL overexpression constructs that caused toxicity upon overexpression in WT BY4741 strains, we also performed Ypk1 dosage rescue growth assays. $P_{MET25}$-Ypk1 or $P_{MET25}$-Ypk1(K367A) (kinase dead) plasmids were co-transformed into BY4741 with the toxic SDL plasmid. Transformants were then cultured overnight in SC media containing 2% raffinose and 0.2% sucrose. 10-fold serial dilutions of overnight cultures starting from $OD_{600} = 1.0$ were then made in sterile water and spotted onto SC solid media with 2% galactose (to induce protein expression) or 2% dextrose (no protein expression), in the presence of absence of methionine to drive Ypk1 overexpression. Serially spotted cultures were allowed to grow in the dark at 30°C for 3 days. Plates were then scanned on a flatbed scanner and growth phenotypes were assessed to determine if Ypk1 overexpression could rescue toxicity of overexpression of the SDL protein.

For broth growth assays, exponential phase cultures growing in rich YP (*Burke et al., 2005*) media with 2% dextrose were diluted to $OD_{600} = 0.1$. 100 µl of each culture was placed in a well in a 96 well plate with vehicle or drug at the indicated concentration. Cultures were grown with orbital shaking at 30°C in a Tecan Infinite M-1000 PRO plate reader (Tecan Systems Inc., San Jose, CA) for 24 hr. Absorbance measurements were taken every 15 min. Absorbance values were converted to $OD_{600}$ values using a standard curve of absorbance values of cultures at known $OD_{600}$ taken on the same plate reader.

## Protein purification and in vitro Ypk1 kinase assay

To purify Ypk1-as kinase, pAX50 (2 µ, $P_{GAL1}$-Ypk1-as-TAP, *URA3*) transformed yAM135-A yeasts were diluted to $OD_{600} = 0.125$ in 3 l of SC 2% raffinose 0.2% sucrose and grown shaking at 30°C to mid-exponential phase. Expression was induced for ~18 hr by the addition of 2% galactose. Cells were harvested by centrifugation and frozen in liquid nitrogen. The cells were then lysed cryogenically using Mixer Mill MM301 (Retsch, Düsseldorf, Germany). The lysate was resuspended at 2 ml/g in TAP-B (50 mM Tris-Cl pH 7.5, 200 mM NaCl, 1.5 mM MgOAc, 1 mM DTT, 2 mM NaVO₄, 10 mM NaF, 10 mM Na-PPi, 10 mM β-glycerol phosphate, 1× complete protease inhibitor [Roche, Basel, Switzerland]). The lysate was clarified by centrifugation at 15×*kg* for 20 min. Clarified lysate was then further centrifuged at 100×*kg* for 1 hr and then brought to 0.15% NP-40 using 10% NP-40 detergent stock. Ypk1-as-TAP fusion was then affinity purified from the lysate using IgG-agarose resin (GE Healthcare). The resin was extensively washed with Protease 3C Buffer (50 mM Tris-Cl pH 7.5, 200 mM NaCl, 1.5 mM MgOAc, 1 mM DTT, 0.01% NP-40, 10% Glycerol, 2 mM NaVO₄, 10 mM NaF, 10 mM Na-PPi, 10 mM β-glycerol

phosphate) and then resuspended in 1 ml Protease 3C Buffer. Ypk1-as was eluted by the addition of 80 U Prescission Protease (GE Healthcare, Little Chalfont, UK) for 5 hr at 4°C. Protease 3C was removed by the addition of glutathione-agarose (GE Healthcare).

Putative Ypk1 substrates were expressed as N terminal GST fusions in BL21 *E. coli*. 1 l cultures were grown at 37°C to mid-exponential phase and then induced at room temperature for 4 hr with 0.5 mM IPTG. Cells were harvested by centrifugation and the fusion proteins were purified by affinity chromatography using glutathione-agarose and standard procedures.

For kinase assays 0.25 µg of Ypk1-as kinase was incubated with purified GST-substrate fusion in Kinase Assay Buffer (50 mM Tris-Cl pH 7.5, 200 mM NaCl, 10 mM $MgCl_2$, 0.1 mM EDTA) with 2 µCi [$\gamma$-$^{32}$P]ATP at 30°C in the presence of absence of 10 µM 3-MB-PP1 for 30 min. Reactions were terminated by the addition of SDS/PAGE sample buffer containing 6% SDS followed by boiling for 5 min. Labeled proteins were resolved by SDS/PAGE and analyzed Coomassie blue staining and autoradiography with Phosphorimager plates (Molecular Dynamics, Sunnyvale, CA) on a Typhoon imaging system (GE Healthcare).

## Preparation of cell extracts and immunoblotting

Cell extracts were made by alkaline lysis followed by trichloroacetic acid precipitation as previously described (*Westfall et al., 2008*). To resolve Lag1 and Lac1 phosphorylated species, 15 µl of TCA extract was resolved by SDS-PAGE (8% acrylamide, 35 µM Phos-tag [Wako Chemicals USA, Inc., Richmond, VA], 35 µM $MnCl_2$ at 160 V). The gel was then transferred to nitrocellulose and incubated with primary antibody in Odyssey buffer (Licor Biosciences Inc., Lincoln, NE), washed, and incubated with IRDye680-conjugated anti-mouse IgG (Licor Biosciences) in Odyssey buffer with 0.1% Tween-20 and 0.02% SDS. Blots were imaged using an Odyssey infrared scanner (Licor Biosciences).

Primary antibodies and dilutions used in this study were: 1:1000 mouse anti-HA (Covance Inc., Princeton, NJ), 1:10000 mouse anti-FLAG (Sigma–Aldrich, St. Louis, MO),1:500 mouse anti-GFP (Roche), 1:500 rabbit anti-pSGK (T256) (to recognize Ypk1 phosphorylated at T504) (Santa Cruz Biotechnology, Dallas, TX ) and 1:10000 rabbit anti-Pgk1 (our laboratory).

For phosphatase treatment of cell extracts, TCA extracts were made as above and the precipitated proteins were solubilized in 100 µl Solubilization buffer (50 mM Tris-Cl pH 8.0, 150 mM NaCl, 2% β-mercaptoethanol, 2% SDS). These extracts were then diluted with 900 µl CIP Dilution buffer (50 mM Tris-Cl pH 8.0, 150 mM NaCl, 11.1 mM $MgCl_2$). 150 U calf intestinal phosphatase (New England Biolabs, Ipswich, MA) was then added and incubated for 2 hr at 37°C. Proteins were then TCA precipitated again and resolved by SDS-PAGE as above.

## Analysis of sphingolipid species

Complex sphingolipids were analyzed by thin layer chromatography. Cultures of strains in mid-exponential phase were adjusted to $OD_{600}$ = 1.0 and 2 ml cultures were labeled with 100 µCi of [$^{32}$P] $PO_4^{-3}$ and cells allowed to grow for 3 hr. Lipids were extracted and resolved as previously described (*Hanson and Lester, 1980*; *Momoi et al., 2004*) with minor modifications. The cell pellet was washed twice with 2 ml water and treated with 5% trichloroacetic acid for 20 min on ice. Pellets were extracted twice with 0.75 ml of ethanol/water/diethyl-ether/pyridine/NH$_4$OH (15:15:5:1:0.018) at 60°C for 1 hr. Glycerophospholipids in the extract were hydrolyzed by treating with 0.1 M monomethylamine at 50°C for 1 hr after which the base was neutralized by addition of 12 µl of glacial acetic acid. Lipids were extracted with 1 ml chloroform, 0.5 ml methanol and phases separated with addition of 1 ml water. For some samples 1 ml of 4 N NaCl was used, in order to facilitate the separation of the aqueous and organic phases. The organic layer was dried under vacuum and resuspended in 50 µl of chloroform/methanol/water (16:16:5) and resolved on a silica gel TLC plate with chloform/methanol/4.2 N NH$_4$OH (9:7:2). Radioactivity on the TLC plate was visualized with a Phosphorimager screen and Typhoon imaging system.

LCBs and LCB-1Ps levels were monitored by liquid chromatography-mass spectrometry- Overnight cultures were adjusted to $OD_{600}$ = 0.2 and allowed to grow to an $OD_{600}$ = 1.0. Cell pellets from 10 ml of this culture equivalent to 10 $OD_{600}$ units was used for analysis. C17-sphingosine (Avanti Polar Lipids, Alabaster, AL), (5 nmol) was added to all samples as an internal standard. Lipids were extracted as described above and the final dried lipid extract was dissolved in methanol/water/formic acid (79:20:1), centrifuged to remove insoluble material and an aliquot of this material was injected into the HPLC for analysis.

Lipid extracts were analyzed using an Agilent 1200 liquid chromatograph (LC; Santa Clara, CA) that was connected in-line with an LTQ Orbitrap XL mass spectrometer equipped with an electrospray ionization source (ESI; Thermo Fisher Scientific, Waltham, MA).

The LC was equipped with a C4 analytical column (Viva C4, 150 mm length × 1.0 mm inner diameter, 5 µm particles, 300 Å pores, Restek, Bellefonte, PA) and a 100 µl sample loop. Solvent A was 99.8% water/0.2% formic acid and solvent B was 99.8% methanol/0.2% formic acid (vol/vol). Solvents A and B both contained 5 mM ammonium formate. The sample injection volume was 85 µl (partial loop). The elution program consisted of isocratic flow at 30% B for 2 min, a linear gradient to 65% B over 0.1 min, a linear gradient to 100% B over 4.9 min, isocratic flow at 100% B for 4 min, a linear gradient to 30% B over 0.1 min, and isocratic flow at 30% B for 18.9 min, at a flow rate of 170 µl/min. The column and sample compartments were maintained at 40°C and 4°C, respectively. The injection needle was rinsed with a 1:1 methanol/water (vol/vol) solution after each injection.

The column exit was connected to the ESI probe of the mass spectrometer using PEEK tubing (0.005″ inner diameter × 1/16″ outer diameter, Agilent). Mass spectra were acquired in the positive ion mode over the range m/z = 250 to 1200 using the Orbitrap mass analyzer, in profile format, with a mass resolution setting of 100,000 (at m/z = 400, measured at full width at half-maximum peak height). In the data-dependent mode, the five most intense ions exceeding an intensity threshold of 30,000 counts were selected from each full-scan mass spectrum for tandem mass spectrometry (MS/MS) analysis using pulsed-Q dissociation (PQD). MS/MS spectra were acquired in the positive ion mode using the linear ion trap, in centroid format, with the following parameters: isolation width 3 m/z units, normalized collision energy 40%, and default charge state 1+. A parent mass list was used to preferentially select ions of interest for targeted MS/MS analysis. To avoid the occurrence of redundant MS/MS measurements, real-time dynamic exclusion was enabled to preclude re-selection of previously analyzed precursor ions, using the following parameters: repeat count 2, repeat duration 30 s, exclusion list size 500, exclusion duration 180 s, and exclusion width 0.1 m/z unit. Data were analyzed using Xcalibur software (version 2.0.7 SP1, Thermo) and LIPID MAPS Online Tools (*Fahy et al., 2007*).

Exact masses of precursor ions in positive ion (M + H$^+$) state obtained from LIPID MAPS Online Tools were as follows- phytosphingosine- 318.3003, dihydrosphingosine-302.3053, phytosphingosine-1 phosphate-398.2666 and dihydrosphingosine-1 phosphate- 382.2717. The ions were further confirmed by tandem MS and identification of the fragments; product ions for the phytosphingosine headgroup was 282.3 and dihydrosphigosine headgroup was 266.4.

## In vitro ceramide synthase assay

Yeast expressing 3xFLAG-Lag1 (wild-type and phospho-site mutations) were grown at 30°C to mid-exponential phase and microsomes were prepared as described previously (*Schorling et al., 2001*) and resuspended in B88 buffer (20 mM HEPES-KOH pH 6.8, 150 mM KAc, 5 mM MgOAc, and 250 mM sorbitol). Ceramide synthase was then immunopurified from these microsomes using anti-FLAG agarose (Sigma) as described previously (*Vallée and Riezman, 2005*). Prior to assembling ceramide synthase reactions, a small alioquot of each (15 µl) immunoprecipitate was taken and proteins were resolved by SDS-PAGE and immunoblotted with anti-FLAG to determine the relative amount of immunopurified ceramide synthase in each reaction. Recovered ceramide levels for each reaction were normalized to the amount of ceramide synthase determined by this procedure.

Ceramide synthesis reaction was assembled in B88 buffer with 5 mg/ml BSA, in a total reaction volume of 0.1 ml containing, 47.5 µl anti-FLAG agarose immunoprecipitates, 50 µM phytosphingosine and 0.1 mM of steroyl (C18)- CoA. Reactions were incubated at 30°C for 1 hr. Reactions were terminated by addition of 0.4 ml methanol/CHCl$_3$ (2:1) and further extracted with 0.4 ml CHCl$_3$ and 0.4 ml water. The CHCl$_3$ phase was separated by centrifugation and removed and dried under vacuum. The dried lipids resuspended in methanol/water/formic acid (79/20/1) and an aliquot analyzed by LC-MS as described above. The product from the reaction, C18-phytoceramide was identified by the exact mass of its precursor ion at 584.5612 and its product ion upon fragmentation at 282.3. Additionally, authentic C18-phytoceramide (Matreya, LLC, Pleasant Gap, PA) standard was used to obtain a standard curve in order to establish that values from ceramide synthase reaction were in the linear range of estimation.

## Acknowledgements

This work was supported by an NIH Predoctoral Traineeship GM07232 and a Predoctoral Fellowship from the UC Systemwide Cancer Research Coordinating Committee (to AM) and by NIH R01 Research

Grant GM21841 and Senior Investigator Award 11-0118 from the American Asthma Foundation (to JT). We thank Tony Iavarone in the QB3/Chemistry Mass Spectrometry Facility for expert assistance with the analysis of sphingolipids by mass spectrometry, Howard Riezman for the gift of the FLAG-Lip1 construct, Jonathan Weissman, Martha Cyert, Benjamin Turk, Daniel Klionsky, and Hidetoshi Iida for other strains or plasmids, Jeffrey Liu and Brian Ting for technical assistance, and other members of the Thorner Lab for various reagents, plasmids, and many helpful discussions.

## Additional information

### Funding

| Funder | Grant reference number | Author |
| --- | --- | --- |
| National Institute of General Medical Sciences | GM21841 | Jeremy Thorner |
| American Asthma Foundation | Senior Investigator Award 11-0118 | Jeremy Thorner |
| University of California | Cancer Research Coordinating Committee (CRCC) | Alexander Muir |

The funders had no role in study design, data collection and interpretation, or the decision to submit the work for publication.

### Author contributions

AM, SR, FMR, Conception and design, Acquisition of data, Analysis and interpretation of data, Drafting or revising the article; GT, Contributed in a major way to screening all of the bioinformatic hits using the synthetic dosage lethal method devised in this paper, Acquisition of data; JT, Conception and design, Analysis and interpretation of data, Drafting or revising the article

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
