## [Decision Letter]

Thank you for sending your work entitled “TORC2-dependent protein kinase Ypk1 phosphorylates ceramide synthase to stimulate synthesis of complex sphingolipids” for consideration at *eLife*. Your article has been favorably evaluated by Tony Hunter (Senior Editor), who acted as the Reviewing Editor, and 3 reviewers.

The Reviewing Editor and the other reviewers discussed their comments before we reached a decision, and the Reviewing Editor has assembled the following comments to help you prepare a revised submission.

They all agreed that this is a rather complete story that illustrates the power of using a combination of protein kinase consensus sequence based bioinformatics and synthetic dosage lethality in yeast to identify protein kinase targets in key cellular pathways. Your finding that stress-activated Ypk1, an orthologue of the mammalian Sgk1 protein kinase, phosphorylates and stimulates the Lag1/Lac1 catalytic subunits of the ceramide synthase complex thus coordinating regulation of the sphingolipid biosynthesis pathway in response to stress is a significant advance and constitutes a very nice piece of work that merits publication in *eLife*. As you may be aware, *eLife* editorial practice is to assemble a decision letter that summarizes the main points that need to be addressed before the paper can be accepted. In this case, the only major point is that better evidence is needed to establish your conclusion that constitutively active Ypk1 is deleterious to growth because it results in induction of autophagy (Reviewers 2 and 3). Otherwise, the reviewers' comments are relatively minor, but appear to be eminently doable and worth addressing in revision, and they have been included below (together with the complimentary prefatory remarks!).

*Reviewer 1*:

The paper describes a unique, well conceived and well described candidate genetic screen to identify phosphorylation substrates of yeast, Ypk1, a TORC dependent protein kinase. Beyond the screen itself, the paper focuses on characterization of the functional significance of two clearly identified targets, Lag1 and Lac1. Most significantly, the study demonstrates that phosphorylation of ceramide synthase by Ypk1 leads to up-regulation of its activity. Moreover, they show that calcineurin dephosphorylates ceramide synthase. These results significantly expand the known targets of Ypk1 and provide significant new details of regulation of sphingolipid synthesis by a conserved protein kinase. This paper constitutes a significant advance in understanding regulation of crucial aspects of sphingolipid lipid synthesis in the yeast model system.

The only significant criticism concerns the data in Figure 6. This part of the Figure is not clearly labeled. Are the two side-by-side lanes, under each strain, repeats? They do not look very reproducible and there is no quantification given in the Table or elsewhere in the paper. Furthermore, there is an unlabeled band below the MIPCs. Is this an unknown lipid or is it M(IP)2C? Where is M(IP)2C? Some quantification of relative amounts of these sphingolipids and their relative proportions in the three strains should be presented and/or discussed in more detail.

*Reviewer 2*:

This is an interesting study combining bioinformatics, genetics, and biochemistry to identify novel substrates of the yeast kinase YPK1 (ortholog of mammalian SGK isoforms). Two highly related ER transmembrane proteins that comprise the ceramide synthase complex, Lac1 and Lag1, are identified as bona fide YPK1 substrates whose phosphorylation is required for survival following inhibition of sphingolipid synthesis, a condition which activates YPK1. Metabolite profiles and enzymatic assays suggest that the phosphorylation stimulates ceramide synthase activity. However, links back to the stresses that activate TORC2 are not sufficiently made. Genetic evidence is provided that the YPK1-mediated stimulation of Lac1/Lag1 protects cells from the growth inhibitory effects of long chain bases and autophagy. This is a particularly interesting aspect of the proposed model, but the link to autophagy is not well established by the data shown.

Specific comments:

1) The expression levels of the various phospho-mutant and mimetic alleles relative to wild-type Lac1 and Lag1 need to be shown under the different experimental conditions. These are stated as “data not shown”, but are essential controls for interpreting the data presented. This is particularly important for the IP-enzymatic assay in Figure 6. The levels of each in the IPs should be shown, with the reactions normalized to differences in enzyme levels in the IP. It is unclear from the current description how these assays were normalized (e.g., what is the “protein unit” on the Y-axis?).

2) To make a clearer link between the stresses that activate YPK1 and the phosphorylation-dependent activation of ceramide synthase, the assays in Figure 6 should be repeated, comparing wild-type to ypk1-deleted strains in the presence or absence of myriocin. Furthermore, the ability of myriocin to stimulate ceramide synthase should be assessed comparing wild-type to the Lac1/Lag1-SSAA mutant.

3) Figure 7: It is proposed that the growth inhibitory effects of YPK1 activation and lc3b-deletion on the Lac1/Lag1-SSAA mutant are through autophagy. This is only supported by an atg1-deletion having very modest rescue effects. It is reasonable that some measurement of autophagy be undertaken in these settings to support such a conclusion.

*Reviewer 3*:

Muir et al. identified a number of novel substrates for Ypk1 kinase using a very elegant genome-wide screen combined with an in vitro Ypk1 kinase assay. Among the hits, the authors demonstrate that Ypk1 directly phosphorylates and activates the ceramide synthases Lag1 and Lac1. This finding answers the long-standing question of how the TORC2-Ypk signaling pathway controls ceramide synthesis, which was previously shown by the Ted Powers group (Aronova et al., 2008). Experiments are properly controlled, the manuscript is well written, and the finding significantly advances our understanding of TORC2-Ypk signaling in sphingolipid metabolism. Specific comments, all relatively minor, are below.

Specific comments:

1) The title “TORC2-dependent protein kinase Ypk1 phosphorylates ceramide synthase to stimulate synthesis of complex sphingolipids” is not fully supported by the experiments. It would be more accurate if the authors demonstrated that Lag1 and Lac1 phosphorylation is regulated also by TORC2, using tor2 or ypk1-T662A (mutated TORC2 target site) mutant cells. Both strains were used in their previous publication (Roelants et al., 2011).

2) Figure 6. The authors demonstrate that Ypk1-mediated phosphorylation of Lag1 and Lac1 stimulates ceramide synthase activity by measuring levels of complex sphingolipids, downstream products of ceramide, rather than by measuring ceramide levels per se. Is there a reason for this?

3) Figure 7. The authors show that lag1/lac1 phospho-deficient mutant (AA) cells exhibit a slow growth phenotype when a constitutively active Ypk1 is expressed. Since an observed upregulation of LCBPs in the AA cells was shown to impede cell growth by triggering autophagy (Zimmermann et al., 2013), the authors conclude the expression of constitutive Ypk1 is deleterious to the growth of the lag1/lac1 AA cells most likely due to hyper accumulation of LCBPs inducing autophagy. The authors should measure if expression of Ypk1 in AA cells indeed increases LCBPs.

4) Figure 7. Related to the point 3 above, it would strengthen the authors' conclusion that Ypk1-mediated stimulation of Lag1 and Lac1 prevents autophagy if they could show autophagy is indeed upregulated upon expression of constitutively active Ypk1 in AA cells.

---

## [Author Response]

*[…] In this case, the only major point is that better evidence is needed to establish your conclusion that constitutively active Ypk1 is deleterious to growth because it results in induction of autophagy (Reviewers 2 and 3). Otherwise, the reviewers' comments are relatively minor, but appear to be eminently doable and worth addressing in revision, and they have been included below (together with the complimentary prefatory remarks!)*.

We appreciate the several suggestions for further improving the impact of this work. As requested by the editor (and reviewers 2 and 3), we now provide new data (by examining processing of GFP-Atg8) demonstrating that basal autophagy is indeed increased when constitutively-activated Ypk1 (which elevates flux into long-chain base synthesis by alleviating Orm inhibition of SPT) is combined with alleles of Lag1 and Lac1 that cannot concomitantly undergo Ypk1-dependent phosphorylation (and thus cannot efficiently divert the long-chain bases generated into ceramides). Moreover, we appreciate 'the complimentary prefatory remarks' from all three referees and have attended to the additional ('relatively minor') issues raised by them, which we now have addressed in all instances by direct experiment, where warranted.

Reviewer 1:

*The paper describes a unique, well conceived and well described candidate genetic screen to identify phosphorylation substrates of yeast, Ypk1, a TORC dependent protein kinase. Beyond the screen itself, the paper focuses on characterization of the functional significance of two clearly identified targets, Lag1 and Lac1. Most significantly, the study demonstrates that phosphorylation of ceramide synthase by Ypk1 leads to up-regulation of its activity. Moreover, they show that calcineurin dephosphorylates ceramide synthase. These results significantly expand the known targets of Ypk1 and provide significant new details of regulation of sphingolipid synthesis by a conserved protein kinase. This paper constitutes a significant advance in understanding regulation of crucial aspects of sphingolipid lipid synthesis in the yeast model system*.

We thank this referee for these extremely laudatory comments.

*The only significant criticism concerns the data in*
Figure 6*. This part of the Figure is not clearly labeled. Are the two side-by-side lanes, under each strain, repeats? They do not look very reproducible and there is no quantification given in the Table or elsewhere in the paper. Furthermore, there is an unlabeled band below the MIPCs. Is this an unknown lipid or is it M(IP)2C? Where is M(IP)2C? Some quantification of relative amounts of these sphingolipids and their relative proportions in the three strains should be presented and/or discussed in more detail*.

In the original Figure 6, and as clearly stated in the legend to original Figure 6, duplicate cultures of each genotype grown to saturation were analyzed; hence, the two side-byside lanes were, in essence, independent repeats. We have done this analysis many times and always see the same trend. However, to address this concern, we repeated this analysis several more times using cultures grown to mid-exponential phase and refined our TLC separation solvent system to better resolve the IPC and MIPC species [we did not resolve MIPCs from M(IP)2Cs]. The IPCs and MIPCs each separate into a family of bands due to differences in the length of the amide-linked fatty acid attached in these ceramides. We now show new representative data for such separations with accompanying quantitation, which fully support our original conclusions.

Reviewer 2:

*This is an interesting study combining bioinformatics, genetics, and biochemistry to identify novel substrates of the yeast kinase YPK1 (ortholog of mammalian SGK isoforms). Two highly related ER transmembrane proteins that comprise the ceramide synthase complex, Lac1 and Lag1, are identified as bona fide YPK1 substrates whose phosphorylation is required for survival following inhibition of sphingolipid synthesis, a condition which activates YPK1. Metabolite profiles and enzymatic assays suggest that the phosphorylation stimulates ceramide synthase activity. However, links back to the stresses that activate TORC2 are not sufficiently made. Genetic evidence is provided that the YPK1-mediated stimulation of Lac1/Lag1 protects cells from the growth inhibitory effects of long chain bases and autophagy. This is a particularly interesting aspect of the proposed model, but the link to autophagy is not well established by the data shown*.

We thank this referee for the generally favorable comments. However, this referee felt that we needed to better document two points: (i) that stimulation of Ypk1 function is linked to TORC2 action; and, (ii) that increased autophagy is linked to elevated long-chain base formation. We now have addressed both by direct experiment. First, using an analog-sensitive Tor2 allele [in cells containing Tor1, Tor2 is only found in the TORC2 complex] and a TORdirected inhibitor (BEZ-235) at 1 μM concentration, we demonstrate that Ypk1-dependent phosphorylation of Lac1 is obliterated almost completely, even in the absence of any stress (see Figure 3, new Panel C). Second, as discussed above in our response to the Editor's comments, we observe a statistically significant increase in the level of GFP-Atg8 processing in Lac1(S23A S24A) Lag1(S23A S24A) cells expressing constitutively-active Ypk1, compared to control cells (see Figure 7, new Panel E).

Specific comments:

*1) The expression levels of the various phospho-mutant and mimetic alleles relative to wild-type Lac1 and Lag1 need to be shown under the different experimental conditions. These are stated as “data not shown”, but are essential controls for interpreting the data presented. This is particularly important for the IP-enzymatic assay in*
Figure 6*. The levels of each in the IPs should be shown, with the reactions normalized to differences in enzyme levels in the IP. It is unclear from the current description how these assays were normalized (e.g., what is the “protein unit” on the Y-axis?)*.

The referee felt that it was important to show, rather than simply state as 'data not shown,' an immunoblot documenting equivalent expression of wild-type Lac1 and Lag1, and the derived Lac1(S23A S24A) Lag1(S23A S24A) and Lac1(S23E S24E) Lag1(S23E S24E) mutants, in the cells in which our genetic and biochemical analyses were performed. A representative example of such data is now provided for one such experiment (see Figure 4, new Panel D), where the control for equal total sample loading is a cytosolic marker protein (Pgk1). In the case of the ceramide synthase assays, the procedure for a comparable immunoblotting analysis to quantify the amount of Lac1 and Lag1 present, and for normalizing to obtain the specific activity, now is given in detail in the revised Materials and Methods and in the accompanying figure legend.

*2) To make a clearer link between the stresses that activate YPK1 and the phosphorylation-dependent activation of ceramide synthase, the assays in*
Figure 6
*should be repeated, comparing wild-type to ypk1-deleted strains in the presence or absence of myriocin. Furthermore, the ability of myriocin to stimulate ceramide synthase should be assessed comparing wild-type to the Lac1/Lag1-SSAA mutant*.

As requested by this referee, we now provide new data (Figure 6, new lower panel) that directly address this issue by showing that, unlike WT cells, which display an increase in ceramide synthase activity in response to challenge with myriocin, otherwise isogenic Lac1(S23A S24A) Lag1(S23A S24A) do not. Moreover, we would like to remind the referee that we already demonstrated that Lac1 (and Lag1) are phospho-proteins *in vivo* (Figure 3), that their phosphorylation *in vivo* requires Ypk1 (Figure 3), that membrane stresses, including myriocin-treatment (Figure 4) and heat treatment (Figure 4), stimulate the Ypk1-mediated phosphorylation of Lac1 and Lag1, and that the inability to phosphorylate the Ypk1 sites in Lac1 and Lag1 causes an increase in LCBs and decrease in ceramides, compared to wild-type cells, determined by two independent methods (Figure 6).

*3)*
Figure 7*: It is proposed that the growth inhibitory effects of YPK1 activation and lc3b-deletion on the Lac1/Lag1-SSAA mutant are through autophagy. This is only supported by an atg1-deletion having very modest rescue effects. It is reasonable that some measurement of autophagy be undertaken in these settings to support such a conclusion*.

As requested by this referee, and as mentioned above in response to the editor, we now provide new data (by examining processing of GFP-Atg8) demonstrating that basal autophagy is indeed elevated both when Lcb3 is absent (so LCB-Ps accumulate) and when constitutively-activated Ypk1 (which elevates flux into long-chain base synthesis by alleviating Orm inhibition of SPT) is combined with alleles of Lag1 and Lac1 that cannot concomitantly undergo Ypk1-dependent phosphorylation (and thus cannot efficiently divert the long-chain bases generated into ceramides, so that LCB-Ps also accumulate) (see Figure 7, new Panel E).

Given the recent report of Zimmermann C et al. (2013) that we cited in the original manuscript, in which it is rather convincingly shown that accumulation of LCB-Ps is a condition that stimulates autophagy, I also think it is fair to state here that the genetic evidence we provided for this conclusion in our case was already quite strong because not only did loss of Lcb3 (phosphatase) make the growth of Lac1(S23A S24A) Lag1(S23A S24A) cells decidedly worse (Figure 7), but conversely loss of Lcb4 (kinase) distinctly ameliorated the growth of Lac1(S23A S24A) Lag1(S23A S24A) cells (Figure 7), in addition to the detectable growth rescue conferred by loss of the essential autophagy inducer, Atg1.

Reviewer 3:

*Muir et al. identified a number of novel substrates for Ypk1 kinase using a very elegant genome-wide screen combined with an in vitro Ypk1 kinase assay. Among the hits, the authors demonstrate that Ypk1 directly phosphorylates and activates the ceramide synthases Lag1 and Lac1. This finding answers the long-standing question of how the TORC2-Ypk signaling pathway controls ceramide synthesis, which was previously shown by the Ted Powers group (Aronova et al., 2008). Experiments are properly controlled, the manuscript is well written, and the finding significantly advances our understanding of TORC2-Ypk signaling in sphingolipid metabolism. Specific comments, all relatively minor, are below*.

We are very gratified that this referee feels that our work 'answers a longstanding question' and also 'significantly advances our understanding' and presents our findings in a manner that is 'properly controlled' and 'well written,' and has concerns categorized as only 'relatively minor.'

*Specific comments*:

*1) The title “TORC2-dependent protein kinase Ypk1 phosphorylates ceramide synthase to stimulate synthesis of complex sphingolipids” is not fully supported by the experiments. It would be more accurate if the authors demonstrated that Lag1 and Lac1 phosphorylation is regulated also by TORC2, using tor2 or ypk1-T662A (mutated TORC2 target site) mutant cells. Both strains were used in their previous publication (Roelants et al., 2011)*.

As recommended by this referee, and already discussed above, we took a different approach, but one highly related to the referee's suggestions, to document that the Ypk1-mediated phosphorylation of ceramide synthase is indeed TORC2-dependent (see Figure 3, new Panel C).

*2)*
Figure 6*. The authors demonstrate that Ypk1-mediated phosphorylation of Lag1 and Lac1 stimulates ceramide synthase activity by measuring levels of complex sphingolipids, downstream products of ceramide, rather than by measuring ceramide levels per se*. *Is there a reason for this?*

In yeast, all of the ultimate, ceramide-derived, sphingolipid pathway products ('complex sphingolipids') contain one or more phosphate groups. Hence, it is experimentally very convenient to measure the rate of *de novo* flux through the entire pathway by measuring incorporation of [32P]PO4 3- into these end-products, which is why we chose to do so.

*3)*
Figure 7*. The authors show that lag1/lac1 phospho-deficient mutant (AA) cells exhibit a slow growth phenotype when a constitutively active Ypk1 is expressed. Since an observed upregulation of LCBPs in the AA cells was shown to impede cell growth by triggering autophagy (Zimmermann et al., 2013), the authors conclude the expression of constitutive Ypk1 is deleterious to the growth of the lag1/lac1 AA cells most likely due to hyper accumulation of LCBPs inducing autophagy. The authors should measure if expression of Ypk1 in AA cells indeed increases LCBPs*.

The experiment requested by this referee has now been performed by mass spectrometry, and expression of constitutively-active Ypk1 in Lac1(S23A S24A) Lag1(S23A S24A) cells does indeed lead to a significance increase in the level of LCB-Ps, compared to the control cells (see Figure 7, new Panel B).

*4)*
Figure 7*. Related to the point 3 above, it would strengthen the authors' conclusion that Ypk1-mediated stimulation of Lag1 and Lac1 prevents autophagy if they could show autophagy is indeed upregulated upon expression of constitutively active Ypk1 in AA cells*.

To corroborate our genetic findings (Figure 7), we now provide biochemical evidence (level of processing of GFP-Atg8) that supports the conclusion that the level of basal autophagy is elevated in Lac1(S23A S24A) Lag1(S23A S24A) cells expressing constitutivelyactive Ypk1, compared to control cells (Figure 7, new Panel E), in agreement with the fact that the same cells exhibit a significant increase in the level of LCB-Ps, compared to control cells (see Figure 7, new Panel B).